# Pathologies of Predictive Diversity in Deep Ensembles

**Taiga Abe**                                                            *ta2507@columbia.edu*
*Center for Theoretical Neuroscience*
*Department of Neuroscience*
*Columbia University*

**E. Kelly Buchanan**                                                    *ekb2154@columbia.edu*
*Center for Theoretical Neuroscience*
*Department of Neuroscience*
*Columbia University*

**Geoff Pleiss**                                                         *geoff.pleiss@stat.ubc.ca*
*Department of Statistics*
*University of British Columbia*
*Vector Institute*

**John Cunningham**                                                      *jpc2181@columbia.edu*
*Department of Statistics*
*Columbia University*

**Reviewed on OpenReview:** *https: // openreview. net/ forum? id= TQfQUksaC8*

## Abstract

Classic results establish that encouraging predictive diversity improves performance in ensembles of low-capacity models, e.g. through bagging or boosting. Here we demonstrate that these intuitions do not apply to high-capacity neural network ensembles (deep ensembles), and in fact the opposite is often true. In a large scale study of nearly 600 neural network classification ensembles, we examine a variety of interventions that trade off component model performance for predictive diversity. While such interventions can improve the performance of small neural network ensembles (in line with standard intuitions), they *harm* the performance of the large neural network ensembles most often used in practice. Surprisingly, we also find that discouraging predictive diversity is often benign in large-network ensembles, fully inverting standard intuitions. Even when diversity-promoting interventions do not sacrifice component model performance (e.g. using heterogeneous architectures and training paradigms), we observe an opportunity cost associated with pursuing increased predictive diversity. Examining over 1000 ensembles, we observe that the performance benefits of diverse architectures/training procedures are easily dwarfed by the benefits of simply using higher-capacity models, despite the fact that such higher capacity models often yield significantly less predictive diversity. Overall, our findings demonstrate that standard intuitions around predictive diversity, originally developed for low-capacity ensembles, do not directly apply to modern high-capacity deep ensembles. This work clarifies fundamental challenges to the goal of improving deep ensembles by making them more diverse, while suggesting an alternative path: simply forming ensembles from ever more powerful (and less diverse) component models.

## 1 Introduction

Successful ensembles require component models that make different errors (Dietterich, 2000; Breiman, 2001; Zhou, 2012; Abe et al., 2022a). This intuition can be formalized by decomposing ensemble performance into

two terms: i) the average performance of component models and ii) the *predictive diversity* across models (see e.g. Breiman, 2001; Webb et al., 2020; Gupta et al., 2022, and Sec. 3). Historically, advances in ensemble methodology have largely focused on the latter term. Consider for example, ensembles of decision trees. Bagged trees (Breiman, 1996) use bootstrapped training data to create different component tree predictions, and random forests (Breiman, 2001) create additional diversity by also subsampling the features used by component models. These training procedures operate at a trade-off; they produce worse component models, but the additional diversity amongst component models yields better aggregate performance (Breiman, 1996; 2001). Random forests generally outperform bagged trees, which outperform single decision trees.

In contrast, deep ensembles (Lakshminarayanan et al., 2017) do not rely on training procedures that actively encourage differences amongst component models. It is instead common to ensemble neural networks with identical architectures and training procedures, where the only source of diversity is the inherent randomness of parameter initialization and stochastic optimization. At first glance, this homogeneity seems massively inefficient—surely we could improve upon deep ensembles if each component model used diverse architectures, training procedures, or dataset/feature subsamples? Surprisingly, despite many proposals to make deep ensembles more diverse—e.g. by varying network architectures, datasets, hyperparameters, or training objectives (e.g. Wenzel et al., 2020; Wu et al., 2021; Gontijo-Lopes et al., 2022; Ortega et al., 2022)—few mechanisms achieve more than a modest gain over the standard deep ensembling benchmark (Ashukha et al., 2020; Abdar et al., 2021; Tran et al., 2022), which remain the gold standard ensembling technique for large scale classification tasks (e.g. Szegedy et al., 2015; Lakshminarayanan et al., 2017; Tran et al., 2022) (see Sec. 2 and Appx. C for a detailed review). More often than not, standard techniques that trade off component model performance for diversity (e.g. bagging) harm ensemble predictions (Lee et al., 2015; Nixon et al., 2020).

This discrepancy between decision tree and deep neural network ensembles should make us reevaluate the notion that encouraging diversity is always a useful technique to improve ensemble performance. On one hand, the relative success of standard deep ensembles might indicate that we have not yet found the right mechanisms to encourage further diversity amongst modern neural networks. On the other hand, it may be the case that intuitions about ensembling simply do not carry over from small decision trees to modern neural networks, which often operate in the overparametrized regime with near zero training error. In other words, the trade-off between component model performance and predictive diversity may be fundamentally different for ensembles of decision trees and deep ensembles.

In this work, we provide strong evidence in favor of the latter hypothesis. We find that for modern classification deep ensembles, prioritizing predictive diversity over component model performance is counterproductive and potentially harmful. We base this claim on two main results. Our first result is concerned with mechanisms that explicitly encourage predictive diversity at the cost of component model performance. We manipulate deep ensemble predictive diversity in a controlled setting via joint-training mechanisms (Mishtal and Arel, 2012; Zhao et al., 2022; Ortega et al., 2022; Abe et al., 2022a). As with many established ensembling techniques, these mechanisms do not encourage predictive diversity "for free," but rather operate at a trade-off: the increased diversity comes at the cost of individual component model performance. Our results, which consider roughly 600 deep ensembles, demonstrate a dichotomy between established ensemble intuitions and the behavior of modern deep ensembles. In particular, encouraging predictive diversity in ensembles of large neural networks like ResNet 18 (He et al., 2016) consistently leads to *worse* ensemble performance, as the increase in predictive diversity between ensemble members does not offset the steep cost incurred on component model performance. This behavior contrasts with that of low-capacity ensembles, including ensembles of small neural networks, which improve under diversity encouragement as expected by standard intuitions. Surprisingly, we also find discouraging diversity does not affect high-capacity deep ensembles, and is sometime beneficial. We attribute these counterintuitive phenomena to (1) the often overlooked effect of diversity encouragement on accurate, confident predictions; and (2) recent evidence around the variance-reducing effects of overparameterization (Adlam and Pennington, 2020; d'Ascoli et al., 2020).

Our second result is concerned with techniques—such as using diverse architectures or training procedures—which increase predictive diversity while maintaining component model performance. While not explicitly harmful, we demonstrate that exploiting such "free" sources of predictive diversity involves an *opportunity cost*. Drawing from a pool of 304 independently trained neural networks, we form over 1000 *homogeneous*

and *heterogeneous* deep ensembles (combining models with the same vs. different architectures) and measure ensemble performance as a function of both predictive diversity and component model performance. In agreement with prior work (e.g. Ashukha et al., 2020; Abdar et al., 2021; Gontijo-Lopes et al., 2022), we indeed find that the "free" predictive diversity of a heterogeneous ensemble yields slightly better performance than homogeneous ensembles. At the same time, our results demonstrate that these slight gains pale in comparison to the benefits of simply using more powerful component models, despite more powerful models being less diverse. In other words, the best deep ensembles are formed by trading off predictive diversity in favor of component model performance, inverting the standard intuition. Critically, this opportunity cost and trade-off depend upon the capacity of component models: for example, we find that while the best modern deep ensembles have very low predictive diversity the best decision tree ensembles have the most predictive diversity. We can observe sharp transitions between these two regimes by ensembling neural network models across a wide range of model capacities.

Taken together, our results strongly suggest that ensembles of modern neural networks operate outside the regime of long-held intuitions around the net benefits of increasing predictive diversity. Moreover, they highlight component model capacity as a critical and overlooked factor affecting ensemble dynamics. Obtaining meaningful predictive diversity that bucks the trade-offs and opportunity costs that we observe will likely require radically new methodology. In the meantime, the best way to improve deep ensembles is simply to use better, but less diverse, individual models.

**Summary of contributions.** We first establish the relationship between our study and previous work in Sec. 2, and describe our experimental setup in Sec. 3. We go on to demonstrate that the size of component models determines the role of predictive diversity in ensemble performance. We show this phenomenon in two distinct settings, focused on large scale classification:

- We show that optimization objectives which encourage or discourage predictive diversity between ensemble members succeed in increasing predictive accuracy on ensembles of small models, but these same objectives fail when applied to ensembles of large component models. We demonstrate this result on over 600 deep ensembles, across various architectures, datasets, and training objectives, and show that our findings reconcile differences between the traditional ensembling literature and various methods proposed to improve deep ensembles (Sec. 4).

- Additionally, we show that our results generalize to many forms of implicit diversity encouragement, i.e. by ensembling across different neural network architectures and hyperparameters. We find that while there may be marginal gains from ensembling more diverse models in such a context, ensembles can be improved more effectively simply by ensembling more powerful, less diverse component models (Sec. 5).

## 2 Related work

**Traditional ensembles.** Historically, ensembling has been applied to low-capacity base models such as decision trees (e.g. Freund, 1995; Breiman, 1996; Dietterich, 1998; Breiman, 2001; Cutler and Zhao, 2001), a notably different setting than modern deep networks. Breiman (2001) identifies the performance of base learners and correlation between ensemble members as the key factors that control ensemble performance. More recent works (Gupta et al., 2022; Wood et al., 2023) discuss diversity metrics that can be derived directly from the methods used to evaluate ensemble member predictions.

**Deep ensembles.** Ensembles of neural networks have long been considered (e.g. Hansen and Salamon, 1990; Perrone and Cooper, 1992), but *deep ensembles* generally refer to ensembles formed over random initialization of modern deep networks. Deep ensembles are most commonly deployed for classification, at times even reformulating regression as classification (Stewart et al., 2022). The standard approach to deep ensembling is to train multiple copies of the same model architecture (e.g. Lee et al., 2015; Lakshminarayanan et al., 2017; Fort et al., 2019; Buchanan et al., 2022; Tran et al., 2022). If available, ensembling different architectures or training procedures yields modest gains in performance (e.g. Fort et al., 2019; Gontijo-Lopes et al., 2022). However actively *encouraging* predictive diversity (i.e., diversity in output space) during training (e.g. Mishtal and Arel, 2012; Lee et al., 2015; Pang et al., 2019; Ross et al., 2020; Thakur et al., 2020; Webb et al., 2020;

Wen et al., 2020; Buschjäger et al., 2020; Jain et al., 2020; Gong et al., 2022; Ortega et al., 2022; Pagliardini et al., 2022; Teney et al., 2022; Zhao et al., 2022) has no consistently demonstrated benefit (see Table 2), except for small networks not often used in practice (Mishtal and Arel, 2012; Opitz et al., 2017; Pang et al., 2019; Webb et al., 2020; Ortega et al., 2022; Pagliardini et al., 2022) or when other training data are available (e.g. Gontijo-Lopes et al., 2022). Beyond acknowledging the failure of existing techniques to improve ensemble performance (e.g. Lee et al., 2015; Nixon et al., 2020) the potentially negative consequences of encouraging predictive diversity in modern deep ensembles are often overlooked.

**Other studies of predictive diversity.** Other works have studied diversity encouragement for logit based averaging of deep ensemble members (Buschjäger et al., 2020; Webb et al., 2020) a setting where the mechanisms proposed in this paper do not necessarily apply. Although recent work suggests that logit averaging is a more theoretically well founded procedure than probability averaging (Gupta et al., 2022; Wood et al., 2023), probability averaging (which is the focus of our study) remains the most frequently used form of deep ensembling (Lakshminarayanan et al., 2017). Two of the methods that we employ to control predictive diversity in deep ensembles were designed to encourage ensemble diversity, as mentioned above (Mishtal and Arel, 2012; Ortega et al., 2022), while the others were originally used to regularize large-model training (Zhao et al., 2022), or to encourage and discourage predictive diversity in regression and certain large scale models (Abe et al., 2022a; Jeffares et al., 2023), respectively. We go beyond the original uses of these methods by systematically studying how diversity encouragement/discouragement differs based on the size of component models across large scale classification tasks. Notably, concurrent work (Jeffares et al., 2023) shows that one of the mechanisms we study here can find degenerate solutions to optimization even in regression tasks or with low-capacity component models. Their conclusions are complementary to our own, and their experiments support the arguments of this work. Finally, we do not address methods which encourage ensemble diversity in the weight space, the feature space, or the gradients of ensemble members (e.g. Dabouei et al., 2020; D'Angelo and Fortuin, 2021; Sinha et al., 2021), and focus on methods which explicitly target diversity in the prediction space of ensemble members. While studying these other sources of ensemble diversity is an interesting point of future work, our primary goal is to highlight the capacity of component models as a critical, understudied feature which determines the impact of encouraging predictive diversity in ensembles.

## 3 The trade-off between predictive diversity and component model performance

**Problem Setting.** We primarily consider multiclass classification problems with inputs $\boldsymbol{x} \in \mathbb{R}^D$ and targets $y \in [1, \ldots, C]$, which is the canonical task for deep neural networks and thus deep ensembles (Hui and Belkin, 2020; Stewart et al., 2022). A standard *deep ensemble* consists of $M$ models $\boldsymbol{f}_1(\cdot), \ldots, \boldsymbol{f}_M(\cdot)$, where each $\boldsymbol{f}_i$ maps input $\boldsymbol{x}$ to the $C$-dimensional probability simplex. The *ensemble prediction* $\bar{\boldsymbol{f}}(\boldsymbol{x})$ is:

$$\bar{\boldsymbol{f}}(\boldsymbol{x}) \triangleq \tfrac{1}{M}\sum_{i=1}^{M}\boldsymbol{f}_i(\boldsymbol{x}). \tag{1}$$

Typically, component models are instances of the same neural network architecture. The component models are usually trained independently on the same dataset with the same loss function $\ell(\boldsymbol{f}_i(\boldsymbol{x}), y)$. This procedure is equivalent to minimizing the average empirical single model risk:

$$\mathcal{L}_{\text{avg}} = \hat{\mathcal{R}}_{\text{avg}} \triangleq \tfrac{1}{M}\sum_{i=1}^{M}\mathbb{E}_{p_{\text{train}}(\boldsymbol{x},y)}\left[\ell\left(\boldsymbol{f}_i(\boldsymbol{x}), y\right)\right] = \mathbb{E}_{p_{\text{train}}(\boldsymbol{x},y)}\left[\tfrac{1}{M}\sum_{i=1}^{M}\ell\left(\boldsymbol{f}_i(\boldsymbol{x}), y\right)\right]. \tag{2}$$

**A decomposition of the ensemble risk.** At test time, we use the aggregated ensemble prediction as in Eq. (1). From Eqs. (1) and (2) we have the following decomposition of the ensemble risk:

$$\mathcal{R}_{\text{ens}} \triangleq \mathbb{E}_{p(\boldsymbol{x},y)}\underbrace{\left[\ell\left(\bar{\boldsymbol{f}}(\boldsymbol{x}), y\right)\right]}_{\text{ens. loss}} = \mathcal{R}_{\text{avg}} - \mathbb{E}_{p(\boldsymbol{x},y)}\underbrace{\left[\tfrac{1}{M}\sum_{i=1}^{M}\ell\left(\boldsymbol{f}_i(\boldsymbol{x}), y\right) - \ell\left(\bar{\boldsymbol{f}}(\boldsymbol{x}), y\right)\right]}_{\text{Jensen gap (pred. diversity)}} \tag{3}$$

(where $\mathcal{R}_{\text{avg}}$ is the non-empirical average single model risk). We assume that $\ell$ is a strictly convex loss (e.g. cross entropy, mean squared error, etc.) and thus the last term—the *Jensen gap*—is non-negative. Throughout this paper we will use the Jensen gap as a measure of predictive diversity (as in Abe et al. (2022a)): it will

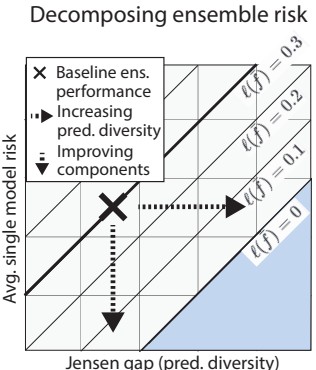

Figure 1: **The diversity/component model performance trade-off**. Ensemble performance, as measured as negative log likelihood (NLL), is decomposed into average single model NLL (vertical axis) and Jensen-gap predictive diversity (horizontal axis)—see Eq. (3). Diagonal lines correspond to level sets of ensemble NLL (lower right is better). The performance of any ensemble can be plotted as a point on this graph ($\times$) with a corresponding *level set* of ensemble performance (thick diagonal line). Along a level set, all ensembles have the same NLL. There are two strategies for improving the performance of any given ensemble: increasing the predictive diversity of the component models (right arrow) or improving the average NLL of the component models (down arrow). If resulting ensembles stay below the thick diagonal, they will improve performance.

be 0 if and only if all component models make the same prediction, and will increase as the $\boldsymbol{f}_i(\boldsymbol{x})$ grow more dissimilar. Thus, Eq. (3) offers a straightforward decomposition of ensemble performance ($\mathcal{R}_{\text{ens}}$) into (i) the average performance of its component models ($\mathcal{R}_{\text{avg}}$) and (ii) the predictive diversity amongst models (Jensen gap), as we will use in our analysis later. Fig. 1 depicts this decomposition: the horizontal and vertical axes correspond to predictive diversity and average component model risk (respectively), while diagonal lines are level sets of ensemble risk. In Appx. D, we verify that the Jensen gap is highly correlated with other metrics of predictive diversity used in previous works.

**Trading off diversity and component model performance.** Fig. 1 captures many of our intuitions around the trade-offs between diversity and component model performance. For example, ensembles of decision trees demonstrate that one can beneficially trade off component model performance in favor of ensemble diversity. Ensembles that improve in this way can be represented in Fig. 1 in the area right of $\times$, below the thick diagonal. Likewise, if encouraging diversity yields *worse* ensemble performance, then it must be due to a loss in the performance of the component models (Fig. 1, area right of $\times$, *above* thick diagonal.) In both cases, we are compromising single model performance to increase ensemble diversity, but there are distinct regimes where this trade-off can help or harm ensemble performance. Note that an analogous statement holds for trade-offs that improve the performance of component models as well: we might hope to improve component models without losing too much diversity (Fig. 1, area left of $\times$, *below* thick diagonal) as an alternative method of improving ensemble performance.

## 4 Manipulating predictive diversity in neural network ensembles

For ensembles of smaller models, methods like bagging and random forests benefit ensemble performance, despite trading off component model performance in favor of predictive diversity. Intuitions from these well-studied paradigms have driven many efforts to achieve this same, favorable trade-off in ensembles of modern deep neural networks. However, in this section we show that this trade-off yields opposite outcomes in modern deep ensembles. In particular, actively encouraging diversity (at the cost of component model performance) is detrimental to deep ensemble performance, while discouraging diversity is benign or beneficial. We find that the degree to which diversity encouragement/discouragement is harmful/benign depends on the capacity of component models. If we substantially decrease the size of component models (e.g. using very small networks like ResNet 8), diversity encouragement once again becomes beneficial—in line with standard intuitions. Nevertheless, we emphasize that the capacity of most modern neural networks used in practice will be in the regime where diversity encouragement is harmful. We attribute this finding to a simple but often overlooked phenomenon: diversity encouragement disproportionately impacts highly confident and accurate predictions, which are a majority of predictions from high-capacity neural networks.

**Mechanisms to encourage/discourage predictive diversity.** To control predictive diversity in these ensembles, we experiment with four modifications of standard ensemble training that enable us to encourage/discourage predictive diversity by a variable amount. The first three mechanisms we consider to

encourage/discourage predictive diversity explicitly regularize standard ensemble training (Eq. 2):

$$\mathcal{L}_{\text{ens.}} = \mathcal{L}_{\text{avg}} - \mathop{\mathbb{E}}_{p_{\text{train}}(\boldsymbol{x}, y)} \Big[ \gamma \underbrace{\mathbb{D}\left(f_1(\boldsymbol{x}), \ldots, f_M(\boldsymbol{x})\right)}_{\text{diversity regularizer}} \Big] \tag{4}$$

where $\gamma$ is a hyperparameter (negative values encourage diversity; positive values discourage diversity) and $\mathbb{D}$ is some measure of predictive diversity.

1. **Variance** regularization (Ortega et al., 2022) sets $\mathbb{D}$ to:

$$\mathbb{D}\left(f_1(\boldsymbol{x}), \ldots, f_M(\boldsymbol{x})\right) = \frac{1}{2(M-1)\max_i f_i^y(\boldsymbol{x})} \sum_{i=1}^{M} \left(f_i^y(\boldsymbol{x}) - \bar{f}_i^y(\boldsymbol{x})\right)^2, \tag{5}$$

which essentially corresponds to the variance over ensemble members' true-class predictions.

2. **1-vs-All Jensen-Shannon divergence**-based regularization (Mishtal and Arel, 2012) sets $\mathbb{D}$ to:

$$\mathbb{D}\left(f_1(\boldsymbol{x}), \ldots, f_M(\boldsymbol{x})\right) = \frac{1}{M} \sum_{i=1}^{M} JS\left(\boldsymbol{f}_i(\boldsymbol{x}) \, \Big\| \, \frac{1}{M-1} \sum_{j \neq i} \boldsymbol{f}_j(\boldsymbol{x})\right), \tag{6}$$

where JS is the Jensen-Shannon divergence.

3. **Uniform Jensen-Shannon divergence**-based regularization (Zhao et al., 2022) sets $\mathbb{D}$ to:

$$\mathbb{D}\left(f_1(\boldsymbol{x}), \ldots, f_M(\boldsymbol{x})\right) = H\left(\bar{\boldsymbol{f}}(\boldsymbol{x})\right) - \frac{1}{M} \sum_{i=1}^{M} H\left(\boldsymbol{f}_i(\boldsymbol{x})\right), \tag{7}$$

where $H$ is Shannon entropy. Note that the term inside the brackets is equivalent to the multi-distribution Jensen-Shannon divergence with uniform weights.

4. **Jensen gap**-based regularization (as introduced in Abe et al. (2022a)) corresponds closely to Eq. (3), as we consider the following objective function to encourage/discourage diversity:

$$\mathcal{L}_{\text{ens}} = \mathop{\mathbb{E}}_{p_{\text{train}}(\boldsymbol{x}, y)} \Big[ \ell\left(\bar{\boldsymbol{f}}(\boldsymbol{x}), y\right) + (\gamma + 1)\left(\frac{1}{M} \sum_{i=1}^{M} \ell\left(\boldsymbol{f}_i(\boldsymbol{x}), y\right) - \ell\left(\bar{\boldsymbol{f}}(\boldsymbol{x}), y\right)\right) \Big]. \tag{8}$$

The first term is the ensemble prediction loss (as opposed to the average single model loss), while the second term is the Jensen gap notion of predictive diversity.

For all methods, $\gamma = 0$ corresponds to standard ensemble training, $\gamma < 0$ encourages diversity, and $\gamma > 0$ discourages diversity. Thus, we can frame encouragement/discouragement of ensemble diversity as regularized versions of standard ensemble training (Eq. 2). As with any regularization mechanism, we expect that manipulating predictive diversity also affects the performance of component models, and we will study how trade-offs between these factors impact overall ensemble performance. Sec. 5 studies the effects of diversity sources that do not fall into this paradigm (diversity using different architectures, training paradigms, etc.) In Appx. F.1, we further discuss differences between these individual diversity mechanisms and evaluate the performance of each regularizer independently.

**Datasets and architectures**. We train deep ensembles on the CIFAR10, CIFAR100 (Krizhevsky et al., 2009), TinyImageNet (Le and Yang, 2015), and ImageNet (Deng et al., 2009) datasets. Ensembles consist of $M = 4$ models of the same neural network architecture for all datasets except ImageNet, where $M = 3$. On CIFAR datasets, we study ResNet 18 (He et al., 2016), VGG 11 (Simonyan and Zisserman, 2014), WideResNet 28-10 (Zagoruyko and Komodakis, 2016), DenseNet 121 (DenseNet 161 for CIFAR100) (Huang et al., 2017), ResNet 8, and LeNet (LeCun et al., 1998) models. The first four are considered large architectures, while the last two are small architectures. Our exemplar large networks are modern neural networks that make highly accurate but overconfident predictions, while our small networks do not (Guo et al., 2017). Additionally, we consider the ResNet 34 architecture for TinyImageNet and the ResNet 18 architecture for ImageNet. In total, we train 574 ensembles across four datasets and six architectures, using the four mechanisms discussed above to encourage/discourage ensemble diversity. (See Appx. B for training and implementation details.)

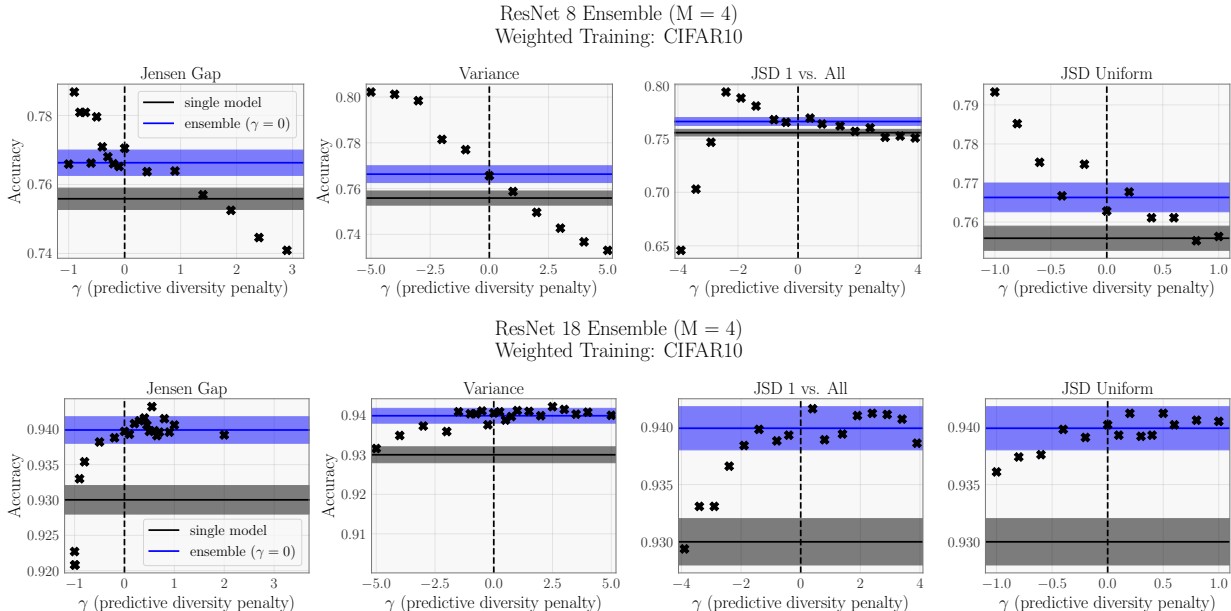

Figure 2: **Encouraging/discouraging diversity has different effects on small vs. large neural network ensembles.** We train deep ensembles of small **(top row)** and large **(bottom row)** neural networks (ResNet 8 and ResNet 18, respectively) with diversity mechanisms on CIFAR10 (one column per diversity mechanism). We compare standard ensemble training (dotted line) to diversity-encouraged/discouraged training (left vs. right of dotted line). Blue lines/bands are standard ensemble test accuracy; black lines/bands are component model test accuracy. Encouraging diversity ($\gamma < 0$) improves test accuracy of small-network ensembles while harming test accuracy of large-network ensembles. Conversely, discouraging diversity ($\gamma > 0$) actively hurts the performance of small-network ensembles, but appears benign for large-network ensembles. (See Appx. F for CIFAR100, TinyImageNet, and other architectures).

| Regularization strength | $\gamma = -0.9$ | $\gamma = -0.5$ | $\gamma = 0$ | $\gamma = 0.1$ |
|---|---|---|---|---|
| Test accuracy (%) | 70.16 | 71.10 | 71.49 | 71.37 |

Table 1: **Diversity encouragement fails on ImageNet.** Diversity encouragement results with ensembles of ResNet 18 models with M = 3, and Jensen Gap regularizer. As with smaller datasets, encouraging predictive diversity has no benefit, while diversity discouragement appears benign.

## 4.1 Results

Within this section, we show results for one representative small and large architecture (ResNet 8 and ResNet 18), with most results on the CIFAR10 dataset. See Appx. F for other architectures and datasets.

**Encouraging diversity benefits small-network ensembles, but harms large-network ensembles.** Fig. 2 (top, $\gamma < 0$) illustrates the effects of encouraging diversity on the accuracy of CIFAR10 ResNet 8 models. (See Appx. F for the other small architecture—LeNet— as well as results on CIFAR100.) Each panel depicts a different diversity encouraging mechanism. The blue line and bands depict standard (unregularized) ensemble performance (mean $\pm$ two standard error), while the black line and bands depict the error of single models. Relative to the standard ensemble baseline (blue band), each diversity-encouraging mechanism yields performance improvements for a range of $\gamma < 0$ (left of baseline). Across all diversity mechanisms, datasets, and model architectures, diversity encouragement improves the performance of 73% of 86 small ensembles (see Appx. F for the other small architecture—LeNet—as well as results on CIFAR100.)

In stark contrast, Fig. 2 (bottom, $\gamma < 0$) illustrates that diversity encouragement has the opposite effect on ensembles of larger ResNet 18 models. For all mechanisms, datasets and model architectures, diversity

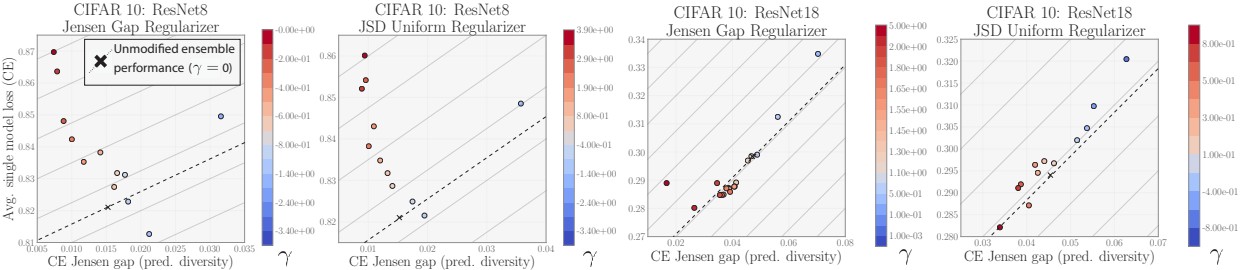

Figure 3: **Trading off predictive diversity and component model performance in diversity regularized ensembles.** Each marker represents a ResNet 8 (**left panels**) or ResNet 18 (**right panels**) ensemble trained with a diversity intervention on CIFAR10 (see Fig. 12 for CIFAR100). Warmer colors correspond to positive $\gamma$ values (encouraging diversity), cooler colors correspond to negative $\gamma$ values (discouraging diversity). Axes are given by ensemble loss decomposition as in Fig. 1. The level set of standard deep ensemble performance ($\gamma = 0$) is denoted by $\times$ and the dotted diagonal line. In all ensembles, encouraging/discouraging predictive diversity leads to proportionally higher/lower diversity on test data. Small-network ensembles with diversity encouragement ($\gamma < 0$, blue markers), can achieve higher single model performance, and thus better ensembles (below dotted line). For large-network ensembles however, diversity encouragement comes at a high cost to average single model performance, and worse ensembles overall (above dotted line). For small-network ensembles, discouraging predictive diversity ($\gamma > 0$; red markers) also leads to worse performance of component models: thus corresponding ensemble performance is also worse (above dotted line) than standard ensembling. In contrast, diversity discouraged large-network ensembles can outperform standard training (below the dotted line). Among large-network ensembles, the one with the best test NLL is one where diversity was *discouraged*. In Fig. 13, we also study the relationship between diversity and average accuracy of these ensembles.

encouragement ($\gamma < 0$) yields worse performance than a standard ensemble in 61% of 134 large networks (at times undershooting single networks), and no statistically significant changes to an additional 29%. These results also hold on large datasets: training ensembles of ResNet 18 on the full ImageNet dataset (Table 1), we find that diversity encouragement follows the same trends as on CIFAR10. (See Appx. F for other large architectures—VGG, WideResNet, and DenseNet—as well as results on CIFAR100 and TinyImageNet.)

We next verify that training with diversity encouragement leads to higher diversity on test data. Fig. 3 depicts the performance of ensembles formed from ResNet 8 models (left panels) and ResNet 18 models (right panels), using the decomposition shown in Fig. 1. We show results on the CIFAR10 dataset for the Jensen Gap and JSD Uniform regularizer. Relative to standard ensemble training (indicated with $\times$), ensembles trained with more diversity encouragement ($\gamma < 0$, blue dots) express proportionally more predictive diversity (further right). Factoring in the average performance of component models provides an alternative quantification of the same trade-off we observe in Fig. 2. While ensembles of small models can benefit from diversity encouragement, (Fig. 3, blue points below the dotted line), for large-network ensembles the effect of encouraging predictive diversity is higher values of the ensemble loss (blue points above dotted line). In Appx. F.2 we show additional loss decompositions for ResNet 18 diversity regularized ensembles on CIFAR100.

**Discouraging predictive diversity is benign for large-network ensembles, but harmful for small-network ensembles.** For the small-network ResNet 8 ensembles (Fig. 2, top, $\gamma > 0$), we observe that all diversity discouraging mechanisms harm test set error, consistent with our intuitions from traditional ensembles. Across 82 small-network experiments (here in Fig. 2 and in Appx. F), diversity discouragement hurt performance 56% of the time, whereas performance remains intact in 40%, and helps for only 4%. In contrast Fig. 2 (bottom, $\gamma > 0$) shows that penalizing diversity does not harm the large-network ResNet 18 ensembles. Across 149 experiments (depicted in Fig. 2 and Appx. F), 74% of large-network ensembles obtain similar or better performance with diversity discouragement (again, see Appx. F for additional results.)

Once again, we verify that discouraging predictive diversity ($\gamma > 0$, red dots), produces ensembles that are less diverse on test data in Fig. 3. In line with the results on predictive accuracy, small model ensembles demonstrate strictly worse ensemble performance when discouraging predictive diversity (red points above the dotted line). For large-network ensembles however, diversity discouragement can sometimes lead to improved ensemble performance (red dots below dotted line). Appx. F.2 shows additional loss decompositions for ResNet 18 diversity regularized ensembles on CIFAR100.

**Similar trends hold for probabilistic performance metrics and calibration.** In Appx. F.5 we replicate Fig. 2 for the negative log likelihood (NLL), Brier Score (BS), and expected calibration error (ECE) (Naeini et al., 2015) metrics measured on the test set. We observe similar trends: encouraging diversity helps/hurts small/large-network ensembles, and vice versa for diversity discouragement.

**Similar trends hold for out of distribution accuracy.** In Appx. D we apply the ensembles from Fig. 2 to CIFAR10.1 (Recht et al., 2018), an out of distribution (OOD) test dataset for CIFAR10. Our results show that OOD performance follows the same trends as InD data.

**Similar trends hold for MLPs on tabular datasets.** Finally, we show that our diversity regularization results generalize to other datasets and network architectures: namely the application of MLPs to tabular datasets (see Appx. B.2.2 for implementation details, and Appx. D.1 for results). Interestingly, in this setting we also discovered models of intermediate capacity that appear to interpolate between the results for the large vs. small model ensembles we consider here.

## 4.2   Analysis

Having verified that the diversity mechanisms studied above successfully encourage/discourage test set predictive diversity, we must conclude that small vs. large neural network ensembles trade off component model performance in different ways (helping vs. harming ensemble performance, respectively). At a high level, we claim that encouraging predictive diversity at the cost of component model performance harms large-network deep ensembles because most of their predictions are confident and correct. In particular, encouraging diversity during training is unlikely to impact only erroneous predictions at test time, as diversity encouragement techniques would then constitute oracle error detectors. Unfortunately, increased predictive diversity is strictly harmful for any correct and confident output, as ensemble risk is only minimized if all models produce the same correct/confident output. This pathology does not affect small-network ensembles, which tend to produce a greater proportion of erroneous/unconfident predictions that can be improved through decorrelation. We provide evidence for these claims below.

**Encouraging diversity during training impacts all predictions, even correct ones.** Fig. 4 studies if diversity encouragement is more likely to impact predictions that would otherwise be correct or incorrect. We first measure the diversity encouraged on each datapoint in the test set using ensembles trained with values of $\gamma \leq 0$. We plot the resulting distribution of diversity values across the CIFAR10 test set for ResNet 8 (left) and ResNet 18 (center) ensembles trained with Jensen-gap diversity encouragement (top row), with different lines representing densities for different values of $\gamma$. Relative to standard ($\gamma = 0$) training, for ensembles trained with diversity encouragement ($\gamma < 0$), there is more data in the rightmost portion of the distribution, demonstrating the impact of diversity encouragement on individual predictions. For each of these predictions, we then consider the predictive accuracy of a standard ($\gamma = 0$) ensemble, which we call the *counterfactual accuracy* (bottom row)- i.e., the accuracy of that prediction in the absence of any diversity encouragement. Ideally, we would want ensemble predictions where diversity is encouraged to have low counterfactual accuracy (i.e. we are diversifying predictions which would otherwise result in errors), but this is not what we see in practice. For ResNet 8 (left) and ResNet 18 (center) ensembles trained with Jensen-gap diversity encouragement, we observe that for interventions which incur greater levels of predictive diversity (lighter colored lines), a *greater* portion of diverse predictions (further to the right) are counterfactually correct. In other words, most datapoints where the ensemble expresses increased diversity *would already have been classified correctly* in the absence of diversity interventions. If diversity encouragement only decorrelated errors, we would expect the opposite: *higher* values of counterfactual ensemble accuracy at low values of predictive diversity, and *lower* values on data that expresses more diversity relative to standard ensembling. It is worth noting that this trend holds for both small- and large-network deep ensembles. Moreover, this

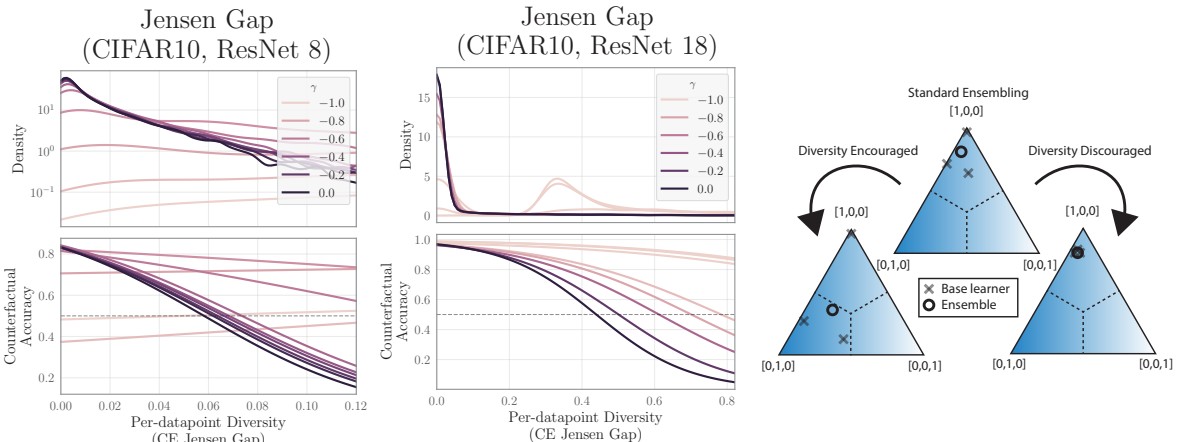

Figure 4: **Diversity encouragement hurts confident/accurate classifier ensembles.** (**Left, center:**) we study the per-datapoint impact of diversity encouragement via *counterfactual accuracy:* the per-datapoint accuracy given by a standard ($\gamma = 0$) ensemble predictions (see Sec. 4.2 for details). Evaluating an ensemble trained with diversity encouragement (Eq. (8)) produces a distribution of ensemble diversity over test set predictions (**top row**, one line per $\gamma$ value). Across this distribution, we then measure the counterfactual accuracy (**bottom row**), and focus on data in the right tail, which are the most strongly influenced by diversity encouragement techniques. Unfortunately, we see that predictions which are most influenced by diversity encouragement have high counterfactual accuracy: i.e., if we were to test a standard ensemble on these datapoints, it *would have been correct anyway*. This finding holds for small (ResNet 8, **left**) and large-network ensembles (ResNet 18, **center**) (for other diversity-encouraging mechanisms, see Appx. I.) (**Right:**) Decorrelating correct/confident component model predictions yields worse ensemble performance. Correct/confident component model predictions ($\times$) concentrate in a vertex of the probability simplex (**top panel**). Due to simplex geometry, encouraging diversity (**left panel**) necessarily degrades the the ensemble prediction ($\circ$), potentially altering the class prediction. In contrast, diversity discouragement (**right panel**) need not hurt ensemble predictions.

phenomenon generalizes beyond the diversity-encouraging mechanisms studied in this section. In Appx. I we demonstrate a similar trend in random forests, suggesting that even traditional ensembling techniques like feature-subsetting impact otherwise correct predictions.

**Predictive diversity is detrimental for otherwise correct/confident predictions.** The above results has significant implications when ensemble members already make highly accurate and confident predictions, as large classification neural networks often do (Guo et al., 2017). In this case, component model predictions will all concentrate in the same vertex of the probability simplex (see Fig. 4, "Standard Ensembling"), and it is impossible to increase predictive diversity without harming the ensemble risk in Eq. (3). Increasing diversity would necessarily require at least one component model to become less confident, reducing the confidence of (and potentially changing) the ensemble prediction (see Fig. 4 right for an illustration).

These claims describe a mechanism to relate observed differences in the impact of diversity encouragement directly to component model capacity. Small neural networks, decision trees, and other low-capacity models produce fewer confident/correct predictions where predictive diversity can only harm component model performance. For the unconfident/erroneous predictions that remain, more diversity will not necessarily harm component model performance, leading to a more favorable trade-off. As a first test of these claims, we would predict that regression ensembles of any size are also immune to this pathology because the output space is unbounded, so underestimated predictions from one component model can cancel out overestimates from another. In Appx. B.2.3, we indeed confirm that large regression deep ensembles can benefit from diversity encouragement, unlike their classification counterparts. Analogously, models trained with a loss that discourages confident predictions—such as a label smoothing loss (Szegedy et al., 2016)—are less susceptible to this pathology, lending additional support for these claims (Appx. D.4).

**Why is discouraging predictive diversity beneficial?** While the previous claims support our diversity-encouragement results, they do not explain why discouraging diversity in large-network deep ensembles is benign (and occasionally beneficial). Here we offer one possible hypothesis for our observed results. Recent theoretical evidence suggests that increasing the size of overparameterized models amounts to variance reduction (Neal et al., 2018; Adlam and Pennington, 2020; d'Ascoli et al., 2020)—a result which is supported empirically by Fig. 6 and other work on ensemble distillation (Lan et al., 2018; Zhao et al., 2022). Discouraging diversity through the use of a regularizer could thus be an alternative path towards variance reduction that mimics the effects of larger models. Confirming this hypothesis is an open problem we leave for future work.

**Relation to existing work.** Importantly, as noted in Sec. 2 and Appx. C we find no evidence in the extensive literature on ensemble diversity mechanisms that contradicts our claims. In particular, across many different mechanisms designed to increase predictive diversity, significant gains to ensemble performance are only reported when applied to ensembles of small networks, or when using other sources of training data, and not when training deep ensembles composed of large neural networks used in practice.

## 5 The best deep ensembles express low predictive diversity

Our results thus far consider methods where there is an explicit cost to increasing ensemble diversity: trading off the performance of component models. There are however sources of ensemble diversity which do not have such a cost, such as ensembling over heterogeneous architectures and training methods. In such cases it is possible to construct diverse ensemble members "for free" by considering component models with different training/architectures but similar performance. Even in such cases, we demonstrate that there is an opportunity cost associated with selecting ensemble members to increase such "free" diversity. While there is substantial predictive diversity amongst low-performance architectures (e.g. AlexNet, VGG 11, etc.) higher-performance architectures (e.g. ResNet 152, DenseNet 161, etc.) tend to produce very similar predictions across datapoints. In other words, ensembling heterogeneous architectures offers diminishing diversity as we consider ensembles of larger component neural networks. This finding is confirmed by prior work, which demonstrates that ensembling diverse architectures/training procedure yields only marginal gains in ensemble accuracy (Gontijo-Lopes et al., 2022). Alternatively, assuming access to a heterogeneous pool of trained models, we could instead construct ensembles that simply use the best component models. Even though the best component models make nearly identical predictions, these ensembles offer far superior performance to those which maximize predictive diversity. These results are a complement to those in Sec. 4: when attempting to form the best ensemble from a pool of candidate models, we find that the best strategy is to trade-off predictive diversity in favor of better component model performance, inverting the standard intuition. (Crucially, we show that this result also distinguishes deep ensembles from traditional small-model ensembles, which are best when the component models are poor but extremely diverse.)

**Experimental setup.** Drawing from a pool of 31 CIFAR10 and 92 ImageNet architectures, we construct *heterogeneous ensembles* of $M = 4$ networks in a combinatorial fashion. We randomly select four independently-trained models that were in the same performance quantile. We additionally construct *homogeneous ensembles* consisting of $M = 4$ independently-trained copies of the same architecture/training procedure. In total, for CIFAR10/ImageNet we construct 1050/427 heterogeneous and 22/3 homogeneous ensembles. Crucially, the models we consider represents a broad swath of existing architectures, including the standard models in Taori et al. (2020). (see Appx. B for architectural/training details). We measure the performance of these ensembles on in-distribution (InD) test sets as well as the CINIC10 (Darlow et al., 2018) and ImageNet-C (Hendrycks and Dietterich, 2019) out-of-distribution (OOD) datasets.

### 5.1 Results

Fig. 5 shows ensemble performance (test-set cross entropy) as a function of average component models' performance. By the decomposition in Eq. (3), the difference between these quantities is the Jensen gap measure of predictive diversity. Models further below the identity line are more diverse, while those closer have little predictive diversity. (See Appx. G for more OOD datasets.)

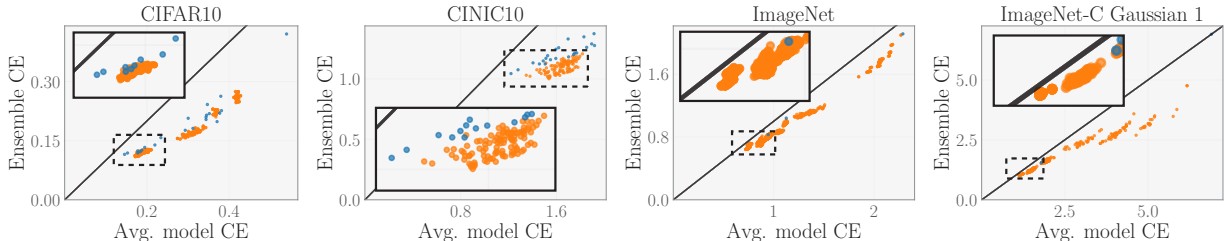

Figure 5: **The opportunity cost of predictive diversity**. Plots depict the ensemble test-set performance versus the performance of its average component model (as measured by cross entropy). Each marker is an ensemble of models evaluated on the InD/OOD datasets of CIFAR10/CINIC10 and ImageNet/ImageNet-C. Dotted boundaries indicate inset region, provided for detail. Ensembles are color coded as homogeneous (blue) or heterogeneous (orange). Controlling for component model performance (vertical slices), heterogeneous ensembles are more diverse than homogeneous ones (further from the identity line). However, heterogeneous ensembles afford diminishing amounts of predictive diversity (closer to identity line) when ensembling higher performance component models (further left). The best performing ensembles (furthest down) have the best component models (furthest left), and very little predictive diversity.

From this plot we observe that conditioning on average component model performance (graphically: consider any vertical slice of Fig. 5), heterogeneous ensembles generally obtain better (lower) performance than homogeneous ensembles. This result suggests that the architectural/training diversity afforded by heterogeneous ensembles is generally beneficial (as expected, since there is no cost to component model performance), confirming the findings of prior work (Mustafa et al., 2020; Gontijo-Lopes et al., 2022; Agostinelli et al., 2022). At the same time, among all heterogeneous ensembles, the best performing ensembles are not the most diverse (i.e., in the bottom right hand corners of Fig. 5 panels). In contrast, the best ensembles are most often formed from the highest performance component models (in the bottom left hand corner of Fig. 5), *even as they become less diverse* (approach identity line). Thus, the diversity gained through heterogeneous architectures/training procedures has diminishing returns as we consider ensembles of more powerful component models, which perform better despite having much lower predictive diversity. Although predictive diversity is not directly harmful to ensemble performance in this case, these results show there is an opportunity cost to forming ensembles to maximize predictive diversity. Counter to classic ensemble intuitions, the best deep ensembles on both InD and OOD datasets in Fig. 5 (the ensembles with the highest average component model performance) are formed by trading off predictive diversity in order to select higher performance component models.

## 5.2 Analysis

Next we study if the opportunity cost we observe in Fig. 5 is also a function of component model capacity. Comparing the behavior of deep ensembles with random forests and bagged random feature models (Fig. 6), we observe how component model performance vs. predictive diversity contribute to ensemble performance as a function of model size. For random forests of various depth (Fig. 6 left), we plot the average component model performance (test-set cross entropy) versus the ensemble's predictive diversity (Jensen gap). Larger dots correspond to deeper random forests. As in Fig. 1, ensemble performance is given by the diagonal level sets. In direct contrast to deep ensembles, the best random forest (lowest diagonal level set) is the most diverse (furthest right).

We next study ensembles of random feature classifiers trained on MNIST. In this setting we can map small vs. large size networks explicitly onto a notion of model capacity: to underparametrized (incapable of fitting training data perfectly) or overparameterized regimes (achieving 100% training accuracy), depending on the classifier width (i.e. the number of random features). In Fig. 6 (center left), we again plot component model performance versus diversity, with diagonal level sets depicting ensemble performance. As we increase the width of the component classifiers (graphically depicted by dot size) we identify over and underparametrized regimes by looking at only average single model performance (Fig. 6 center left, y-axis) as a function of

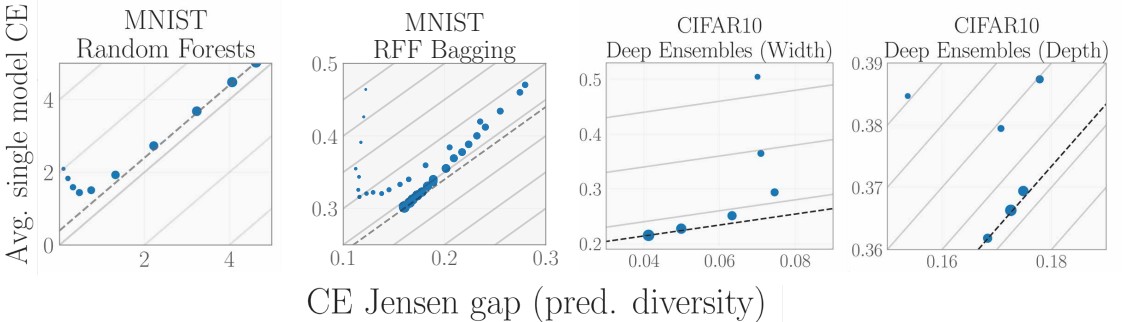

Figure 6: **Small vs. large ensembles improve performance through different strategies**. Ensembles of decision trees (**left**), random feature classifiers (**center left**), and neural networks (**center right/right**). Dotted diagonal line denote the best ensemble cross entropy achieved within a given model class (lower right is better). Dot size corresponds to depth/width of the component models. **(Left):** Random forests improve with increasing diversity (obtained by using deeper component trees). **(Center Left):** Random feature ensembles monotonically improve with classifier width. When component models are underparameterized (smaller dots), ensemble improvement corresponds to increased predictive diversity. When component models are overparameterized (larger dots), ensemble improvement corresponds to decreased predictive diversity. **(Center Right/Right):** Deep ensembles monotonically improve with width/depth. After a certain size, this improvement is entirely attributed to better component models with less predictive diversity.

model size (size of markers). As model size increases, average single model performance demonstrates "double descent" (Belkin et al., 2019) with the peak in performance indicating the interpolation threshold. Strikingly, we see that ensemble diversity also appears to change behavior at the interpolation threshold, increasing as a function of model size in the underparametrized regime, and decreasing in the overparametrized regime. While increasing width always improves ensemble performance, this improvement is due to increased diversity in the underparameterized regime, trading off component model performance to achieve this increased diversity, as we see in ensembles of decision trees (Fig. 6 center left; small dots). Meanwhile, it is due to better component classifiers in the overparameterized regime, which trade off predictive diversity for better component models.

We replicate this finding in ensembles of ResNets by varying component model width (Fig. 6 center right) and depth (Fig. 6 right) as proxies for model capacity (see Appx. B.4 for details). Again, increasing width and depth (graphically represented by dot size) yields better ensembles. Ensemble improvement and diversity are initially correlated when networks models are small, yet they become anticorrelated after the component networks grow sufficiently large. We expect that this transition roughly corresponds to under vs. overparametrized ResNet architectures, but the presence of regularization in the course of training prevents a conclusive identification of this transition point. These results demonstrate that the diminishing returns of encouraging predictive diversity are not specific to modern deep ensembles, but rather a general phenomenon in ensembles of large (likely overparametrized) models, which operate in a regime where trading off predictive diversity in favor of component model performance leads to better ensembles.

## 6 Discussion

We demonstrate that encouraging or selecting for predictive diversity when forming deep ensembles (or more generally, ensembles of high-capacity classifiers) is a suboptimal and potentially detrimental strategy. This finding holds due to explicit trade-offs or opportunity costs relating predictive diversity to component model performance. We contrast this behavior to low-capacity neural network ensembles (and regression ensembles), where we observe that seeking increased predictive diversity yields a net benefit to ensemble performance, indicating that our findings are thus entirely consistent with the traditional ensemble literature.

At a high level, our findings demonstrate how valuable intuitions from traditional ensembling settings must be qualified by a notion of component model capacity. Other work has identified the capacity of component

models as an important feature of a deep ensemble. For example, Kondratyuk et al. (2020) and Lobacheva et al. (2020) study the question of when an ensemble is more efficient than a single model of matched parameter count, and Kobayashi et al. (2021) finds that the performance of a deep ensemble degrades to that of a single model as we ensemble models of increasing width. Theisen et al. (2023) provides an alternative characterization of the relationship between predictive diversity and average component model performance in ensembles. Although they do not study the impact of encouraging/seeking predictive diversity in ensemble models, they observe that ensembling becomes less effective when considering models beyond the interpolation threshold, in line with some of our results in Sec. 5. Likewise, theory suggests that many of the benefits of ensembling may simply recapitulate the benefits of increasing width of component models (Adlam and Pennington, 2020; Geiger et al., 2020; Bernstein et al., 2021).

An important goal for further study is to describe the relationship between ensemble member capacity, ensemble member performance, and the phenomena that we observe here in greater detail. Based on our results in Sec. 4, we expect that the difference between regimes where diversity encouragement improves or degrades ensemble performance is due to the proportion and quality (correct vs. incorrect) of predictions that are made with high confidence. While both the performance of individual models and model capacity and has been correlated with high confidence predictions in previous research (e.g. Guo et al., 2017; Nakkiran et al., 2021; Bernstein et al., 2021), it would be useful to know if one (or both) of these factors can predict whether diversity encouragement should help or hurt ensemble performance. We might hope to disentangle these factors by performing diversity encouragement with unregularized models of varying capacity (inspired by Fig. 6 for RFF models) to better characterize the transition between helpful and harmful diversity encouragement across different datasets and regularizers. More broadly our results may be related to observations made in Breiman (1996), which already identify the "stability" of a learning model's predictions (with regard to training data) as a critical feature which determines the success of bagging: unstable predictors can benefit from bagging, but those that generate stable predictions across different training sets will not improve. It would be an interesting point of future work to consider our results within the framework of model stability relative to the various sources of predictive diversity that we consider.

Altogether, our results along with this prior work suggest that as we use larger component models in an ensemble, we should naturally expect the diversity within that ensemble to decrease. Moreover, our results imply that ongoing efforts to introduce *additional* sources of predictive diversity into deep ensembles (through joint training mechanisms, heterogeneous architectures, and similar) will not change this fact, unless they are able to overcome several fundamental challenges, such as encouraging predictive diversity only upon counterfactual errors, and identifying "free" sources of diversity which offer a greater advantage than using more powerful, less diverse models. While ensembling deep neural networks is undoubtedly a practically useful technique to improve performance of neural network models in current use, we expect that meaningful sources of predictive diversity will become even harder to come by as neural network architectures and training paradigms continue to improve.

### Broader Impact Statement

One conclusion of our work is that creating ensembles from more powerful models is a better strategy than attempting to find diverse sets of existing models. Thus, a potential negative outcome of our work could be to further encourage the expenditure of computational resources by training new, larger scale component models instead of reusing existing ones.

At the same time, our work finds few benefits to the use of joint training strategies to create more diverse large capacity deep ensembles, which are computationally expensive in and of themselves. Our hope would be that this conclusion may help to mitigate some of the current computational costs of creating new ensemble models.

### Acknowledgements

We thank John Miller for sharing models trained on CIFAR10, as well as Taori et al. (2020) for making their trained ImageNet models and code open sourced and easy to use.

We would also like to thank Rich Zemel for his thoughtful suggestions, as well as Yixin Wang, Yaniv Yacoby, Zelda Mariet, Dustin Tran and Jeremy Bernstein for their insightful comments, and the Zuckerman Mind Brain Behavior Institute and the Center for Theoretical Neuroscience for additional support.

TA is supported by NIH training grant 2T32NS064929-11. EKB is supported by NIH 5T32NS064929-13 and NIH U19NS107613. JPC is supported by the Simons Foundation, McKnight Foundation, and Grossman Center for the Statistics of Mind. TA, EKB, and JPC are supported by NSF 1707398 Gatsby Charitable Trust GAT3708.

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

## A    Computational resources

The large majority of our experiments were performed on cloud based computational resources (AWS and GCP). Ensembles trained with diversity regularizers in Figs. 2, 3, 6, 12, 14, 15 and 17 to 22 were trained with EC2 instances from the P3 family. We used `p3.xlarge` instances for model training, unless we experienced issues with GPU memory. In these cases we trained models on `p3.8xlarge` instances. We estimate that on average, large model training for CIFAR10 ensembles required 3 hours of compute time, for CIFAR100 ensembles required 5 hours of compute time, and for TinyImageNet ensembles required 8 hours of compute time. We estimate that small model training required 30 minutes per ensemble for all datasets. Collectively, we therefore estimate that these experiments required 1250 compute hours.

Separately, ensembles trained with diversity regularizers in Fig. 8 and Table 1 were trained on 4 x NVIDIA V100 compute instances. We estimate that these experiments required around 700 additional compute hours.

Analysis of experimental results and visualization were performed on M1 MacBook Airs.

## B    Model training

### B.1    Code availability.

The code used to train deep ensembles is provided here: [https://github.com/cellistigs/ensemble_attention/tree/dkl](https://github.com/cellistigs/ensemble_attention/tree/dkl). Additional code for data analysis, as well as training non-deep neural network ensembles is located here: [https://github.com/cellistigs/interp_ensembles](https://github.com/cellistigs/interp_ensembles). Unless otherwise indicated, ensemble training requires a GPU. All model training for ensembles with diversity mechanisms is managed through a single `run.py` script, which takes configuration parameters from the file `ensemble_attention/configs/run_default_gpu.yaml`. Additional parameters, such as learning rate schedule, number of ensemble members, or evaluation on out of distribution data can be managed through this configuration file.

**Classification ensembles with diversity mechanisms** can be trained by running the command:

```
python ensemble_attention/scripts/run.py ++classifier={classifier_name}
++gamma={gamma_value} ++module={ensemble_dkl,ensemble_js_avg,ensemble_js_unif,ensemble_p2b}
```

The provided `module` arguments will train the Jensen gap, JSD 1 vs. All, JSD Uniform, and Variance regularizers, respectively, and ensemble predictions will be saved following training. Classifiers available for training are listed in `ensemble_attention/src/ensemble_attention/module.py`, within the dictionary `all_classifiers`.

**MLP ensembles with diversity mechanisms** can be trained by running the command:

```
python ensemble_attention/scripts/run.py ++classifier={classifier_name}
++gamma={gamma_value} ++module=tabular_ensemble_jgap
```

**Regression ensembles with diversity mechanisms** can be trained with:

```
python ensemble_attention/scripts/run.py ++classifier={classifier_name}
++gamma={gamma_value} ++module=casresgress_ensemble_dkl
```

**Homogeneous deep ensembles** can be trained with the command:

```
python ensemble_attention/scripts/run.py ++classifier={classifier_name}       ++gamma=0
++module=ensemble
```

**Heterogeneous deep ensembles** include independently trained component models. They can be trained with the command:

```
python ensemble_attention/scripts/run.py ++classifier={classifier_name}       ++gamma=0
++module=base
```

**Ensembled resnets of increasing width/depth** can be trained with the command:

```
python interp_ensembles/scripts/run.py ++classifier={classifier_name}          ++gamma=0
++module=ensemble
```

A list of all classifiers can be found at `interp_ensembles/src/interpensembles/module.py`, within the dictionary `all_classifiers`.

**Random forests** can be trained with the command:

```
python interp_ensembles/scripts/trees/rf_large.py
```

This script can be run on an M1 MacBook, and will save probability outputs and corresponding test data labels for trees of all depths.

**Ensembles of random feature models** can be trained with the command:

```
python interp_ensembles/scripts/random_feature_regression.py
```

This script can be run on an M1 Macbook, and will save the labels, logits, and probability outputs of ensemble models trained at each width in separate directories.

## B.2 Diversity regularization experiments.

For all diversity regularization experiments where computing constraints allowed, we calculated two standard error of the mean ensemble accuracy, and quantified if ensembles trained with diversity mechanisms had higher, lower, or equivalent performance relative to this band. We report these quantifications in the main text and in Tables 3 to 6.

### B.2.1 Models in Fig. 2

For a given dataset, all large capacity ensembles and single networks depicted in Figs. 2, 14, 15 and 17 and Table 1 were trained with the same training schedule and hyperparameters for simplicity.

For CIFAR10, we chose 100 epochs of training, with batch size 256, base learning rate $1e - 2$, weight decay $1e - 2$. The optimizer used was SGD with momentum, with a linear warmup of 30 epochs and cosine decay. The best performing checkpoint on validation data was selected after training and used for further evaluation.

For CIFAR100, we ran training for 160 epochs, with batch size 128, base learning rate $1e - 1$, weight decay $5e - 4$. The optimizer used was SGD with momentum, with a tenfold decrease in the learning rate at 60 and 120 epochs. The best performing checkpoint on validation data was selected after training and used for further evaluation.

For TinyImageNet, we used the same training hyperparameters as with CIFAR100, except that training continued for 200 epochs.

For ImageNet models, we ran training for 90 epochs, with batch size 256, base learning rate $1e - 1$, weight decay $1e - 4$. The optimizer used was SGD with momentum, and the learning rate schedule was given as $0.1^{(m/30)}$, where $m$ is the epoch index.

In all experiments, we created ensembles of size ($M = 4$). The SEM bands in Fig. 2 are generated using ($N = 4$) independently trained ensembles or samples. For CIFAR10, we independently trained a new set of ensembles and single models as a baseline; no models are reused between ensembles and single models in Fig. 2, and ensembles of size $M = 4$ are created by subselecting models from a collection of 5 independently trained models. For CIFAR100 and TinyImageNet, and in Figs. 14 and 15, we create ensemble SEM bands by considering all of the $\gamma = 0$ seeds we trained across different regularizers. For ImageNet, we do not create these bands due to computational constraints.

Small-capacity models are trained using the same losses and training schedule as their large capacity counterparts in Fig. 2, but we choose to keep the CIFAR10 learning rate and hyperparameters for all models and datasets in this set of experiments.

### B.2.2    Models in Fig. 7 (MLPs on tabular datasets)

To find a reasonable set of model hyperparameters for the covertype dataset, we first ran hyperparameter optimization on the number of layers, layer size, learning rate, and weight decay of MLPs using optuna and following parameter range suggestions in Gorishniy et al. (2022). We then fixed the learning rate and weight decay ($5e-4, 2e-5$) from this hyperparameter sweep, and considered 4 different model architectures: **smaller** (2 layers, 32 neurons per layer), **small** (2 layers, 64 neurons per layer, **big** (8 layers, 512 neurons per layer), **bigger** (8 layers, 1024 neurons per layer). Models were trained for a maximum of 100 epochs using the AdamW optimizer, with early stopping.

### B.2.3    Models in Fig. 8 (diversity mechanisms for models trained with MSE loss)

The models in this section are trained using a Mean Squared Error (MSE) loss instead of a softmax cross entropy loss following Hui and Belkin (2020); Demirkaya et al. (2020). The ensembles are trained using WideResNet 28-10 architectures (Zagoruyko and Komodakis, 2016) and DenseNet 161 architectures (Huang et al., 2017) for both the CIFAR10 and CIFAR100 (coarse) datasets (Hui and Belkin, 2020). The implementation of WideResNet 28-10 is from the open source PyTorch implementation in Zagoruyko and Komodakis (2016).

The models trained on CIFAR100 are trained using coarse labels, where the number of classes is 20 instead of 100, as deep neural networks do not achieve good performance for datasets with a large number of classes in this setting Hui and Belkin (2020); Demirkaya et al. (2020).

### B.3    Standard ensemble models in Figs. 5 and 6

**MNIST models.** We fit random forests to MNIST using standard implementations available in scikit-learn (Pedregosa et al., 2011), with the maximum proportion of features sampled by an individual tree set to 0.7- all other parameters were set to defaults. In order to obtain non-zero probability estimates, we pad all probabilistic outputs with a small constant, $\epsilon = 1^{-10}$, and renormalize probabilities.

Additionally, we train random feature models on MNIST as well. We construct random fourier feature models, whereby any data vector $\vec{x} \in \mathbb{R}^N$ is projected to $\cos(\mathbf{W}\vec{x} + b)$, where $\mathbf{W} \in \mathbb{R}^{D \times N}$, where $D$ is the width of the model. Here $\mathbf{W} \sim \mathcal{N}(0, \frac{1}{D})$ and $b \sim \text{Unif}[0, 2\pi]$. We fit the resulting features via logistic regression using standard implementations available in scikit-learn (Pedregosa et al., 2011), combined with the `BaggingClassifer` to generate bagged ensembles.

**CIFAR models.** The models used to generate these plots include three subsets of ensemble types. The first subset consists of models that were trained with the same hyperparameters as in Appx. B.2.1. These models are all of size $M = 5$, of the following architectures:

- ResNet 18 He et al. (2016)

- WideResNet 18-2, 18-4, 28-10 Zagoruyko and Komodakis (2016)

- GoogleNet, Inception v3 Szegedy et al. (2015)

- VGG with 11 and 19 layers Simonyan and Zisserman (2014)

- DenseNet 121 and 169 Huang et al. (2017)

The second subset consists of model implementations derived from the code released with Zhao et al. (2022), found here: `https://github.com/FreeformRobotics/Divide-and-Co-training`. These ensembles of larger models are of the following architectures:

- Pyramidnet 110 (M=4) Han et al. (2017)

- SeResNet 164 (M=5) Hu et al. (2018)

- ShakeShake 26 2x96d (M=5) Gastaldi (2017)

- ResNexst 50 4x16d (M=5) Xie et al. (2017); Zhang et al. (2022); Zhao et al. (2022)

For these models, we changed the learning rate to $1e - 1$, and the weight decay to $1e - 4$.

Finally, we used 36 separate sets of ensembles from Miller et al. (2021), constituting 137 individual models, and we thank the authors for graciously sharing these results with us. Altogether, our results encompass 206 individual deep networks.

To generate heterogeneous ensembles, we first recorded the distribution of ensemble sizes among these separate subsets of ensembles. We then randomly shuffled all of our individual models together, and divided them into new deep ensembles, respecting the distribution of ensemble sizes seen in the original set of homogeneous ensembles. We computed all metrics of interest on 10 random shuffles of ensemble members, and plotted the spread across those random shuffles as the orange dots in Fig. 6.

**ImageNet models.** The 98 models trained on ImageNet come from Abe et al. (2022b); Taori et al. (2020), which are publically available. We provide additional experimental details on how we generated heterogeneous ensembles from these models in the main text.

### B.4 ResNets of increasing width/depth in Fig. 6

In Fig. 6, right hand panels, we consider ensembles of ResNet models with increasing width or depth and examine how these ensembles improve in terms of predictive diversity and average single model performance.

**Increasing width.** For models of increasing width, we consider as our basic architecture the "ImageNet style" ResNet 18 architecture from He et al. (2016), applied to CIFAR10 (see https://github.com/huyvnphan/PyTorch_CIFAR10 for an implementation). Relative to this standard implementation, we multiply the number of channels in each layer by a factor of $\{\frac{1}{16}, \frac{1}{8}, \frac{1}{4}, \frac{1}{2}, 1, 2, 4\}$ in order to generate networks of varying width. Models were trained with the same hyperparameters as in Appx. B.2.1.

**Increasing depth.** For models of increasing depth, we consider the "CIFAR10 style" ResNet architecture described in He et al. (2016), with three filter sizes and $2n$ layers per filter size for a total of $2n + 2$ layers. We consider $n = \{1, 2, 3, 4, 5, 6, 7\}$ in these experiments, and do not vary training hyperparameters with depth: we ran training for 160 epochs, with batch size 128, base learning rate $1e - 1$, weight decay $5e - 4$, and a learning rate schedule that divided the learning rate by 10 at 60 and 120 epochs.

## C Encouraging diversity during training provides small gains relative to simple deep ensembles

In Table 2, we provide a more detailed summary of various ensemble methods that have been proposed by the community. We contrast *dependent training*, where ensemble members are explicitly trained to be diverse, with *independent training*, where each ensemble member is unaware of other ensemble members during training. While some independent training methods can offer "free diversity" as described in the main text (such as dataset-diverse ensembles), most depedent training mechanisms offer only small benefits to accuracy, although they can improve other metrics of interest such as ECE, or robustness to adversarial attacks. We contrast dependent training mechanisms which explicitly encourage diversity on the training set to those which assume access to some auxiliary dataset (extra data diversity).

## D Additional experiments

In the main text, we study the impact of diversity mechanisms on a subset of small vs. large capacity classification ensembles of convolutional datasets, in terms of standard test error. In this section, we provide evidence from other experiments which also support the conclusions of the main text.

| Method | Dependent Training | | Independent Training | Outperforms baseline? |
| | Train set diversity | Extra Data diversity | | |
|---|---|---|---|---|
| Simple Ensembles Lakshminarayanan et al. (2017) | | | x | Baseline |
| Expert ensembles Mustafa et al. (2020) | | | x | Yes, (but against a finetuning baseline) |
| Ensembles via pretraining Agostinelli et al. (2022) | | | x | Yes (by optimizing pretraining performance) |
| Dataset-diverse Ensembles Gontijo-Lopes et al. (2022) | | | x | Yes, (but via diverging training methodologies/additional training data). |
| JSD Ensembles Mishtal and Arel (2012) | x | | | No (training/inference time gain) |
| TreeNets with MCL loss Lee et al. (2015) | x | | | No (training/inference time gain) |
| DivTrain Kariyappa and Qureshi (2019) | x | | | Yes, (when considering specific adversarial attack) |
| Empirical study* Webb et al. (2020) | x | | | No |
| Batch Ensembles Wen et al. (2020) | x | | | No (training/inference time gain) |
| MIMO Havasi et al. (2021) | x | | | No (inference time gain) |
| Divide and Co-train Zhao et al. (2022) | x | | | Yes, accuracy + < 1% |
| FiLM Ensembles Turkoglu et al. (2022) | x | | | No (training/inference time gain) |
| Empirical study* Ortega et al. (2022) | x | | | No |
| $F_{sum}$ Gong et al. (2022) | x | | | Yes, (improved ECE 1) |
| Diversity Loss Teney et al. (2022) | x | | | Yes, accuracy + < 2% |
| LIT Ross et al. (2020) | | x | | Yes, accuracy + < 1% for 9/10 and + < 3% for 1/10 datasets |
| DICE Rame and Cord (2021) | | x | | Yes, accuracy +2% |
| ERD Tifrea et al. (2022) | | x | | Yes, (for novelty detection) |
| D-BAT Pagliardini et al. (2022) | | x | | Yes, accuracy +1% for 5/6 and < 10% for 1/6 datasets |
| DivDis Lee et al. (2022) | | x | | Yes, (on worst group accuracy) |

Table 2: **Improvements to baseline deep ensembles are difficult to achieve**. Across different training methodologies, we find that improvements to the standard deep ensemble methodology (Lakshminarayanan et al., 2017) are difficult to achieve, unless one assumes access to some additional source of data through which to achieve extra diversity.

## D.1 MLPs

Fig. 7 extends the results in the main text from convolutional neural networks trained on vision classification tasks, to MLPs trained on tabular data. Using the covertype (Blackard, 1998) dataset and the Jensen gap regularizer, we observe that the impact of diversity mechanisms in this setting mirrors our results in the main text. In particular, for large MLP architectures (Fig. 7 top row, "Bigger MLP"), diversity encouragement leads to uniformly worse performance, across the range of encouragement values and performance metrics (accuracy, NLL, Brier Score) considered. However, as we consider slightly smaller architectures (Fig. 7 second row, "Big MLP"), we observe a local optimum in diversity encouragement strength, where performance appears to improve relative to baseline ensemble training, before then declining below the level of standard ensemble training as we further encourage predictive diversity. If we consider an even smaller architecture (Fig. 7 third row, "Small MLP") we see that as a function of regularization strength, encouraging predictive diversity leads to a local maximum in NLL and Brier scores before improving once regularization strength is increased further, thus inverting the behavior of "Big MLP". We do not observe this same correspondence in accuracy, where "Small MLP" appears to display very high variance at all levels of diversity encouragement. Finally, for the smallest MLP architecture we consider (Fig. 7 bottom row, "Smaller MLP"), we recover the performance of small convolutional architectures discussed in the main text: diversity encouragement leads to uniform improvements in ensemble performance. The existence of these local optima in neural network performance is reminiscent of previous work (Webb et al., 2020), that discovered a similar phenomenon when forming ensembles by averaging logits. Such local optima may be related to the behavior of outlier ensembles within the convolutional architecture/vision dataset setting that we focus on (see Tables 3 and 4). Better understanding how such optima in regularization strength evolve as a function of model capacity and ensemble aggregation method will be an important point of future work.

## D.2 Regression deep ensembles

Fig. 8 demonstrates that diversity encouragement in regression ensembles performs similarly to small scale classification ensembles. We plot the predictive performance of a WideResNet 28-10 ensembles (left) and DenseNet 161 ensembles (right) on the CIFAR10 and CIFAR100 coarse datasets, all of which are trained with the MSE loss (rather than cross-entropy) and with diversity encouragement/discouragement using the Jensen Gap regularizer (other regularizers either collapse to the Jensen Gap (Appx. F.2) or are designed for classification specifically). Unlike in Fig. 2, the majority of ensembles trained with diversity encouragement (12/16) improve upon baseline performance, with improvements proportional to regularization strength. In

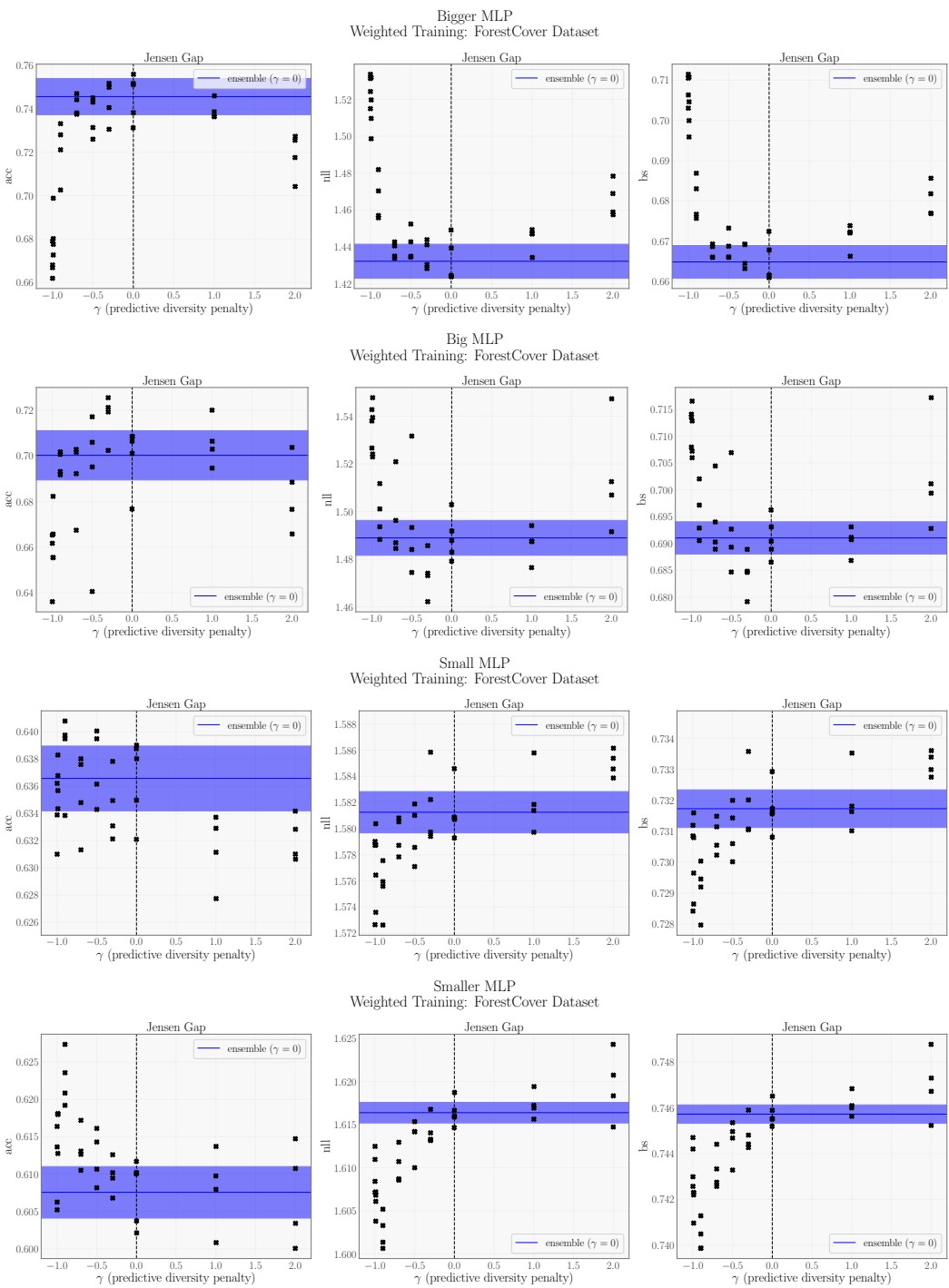

Figure 7: **The effect of diversity mechanisms is dictated by model capacity in MLPs trained on tabular datasets.** We study ensembles aggregating MLPs of different sizes on the covertype (Blackard, 1998) tabular dataset, encouraging and discouraging diversity using the Jensen Gap regularizer. For all ensembles, we report predictive accuracy (**left column**), NLL (**middle column**), and Brier Score (**right column**) (axes as in Fig. 2). Moving from the biggest MLP ensemble that we consider (**top row**) to the smallest (**bottom row**), we see that the impact of diversity encouragement inverts, with diversity encouragement helping/ hurting for smaller/larger model ensembles, respectively. See Appx. D.1 for details.

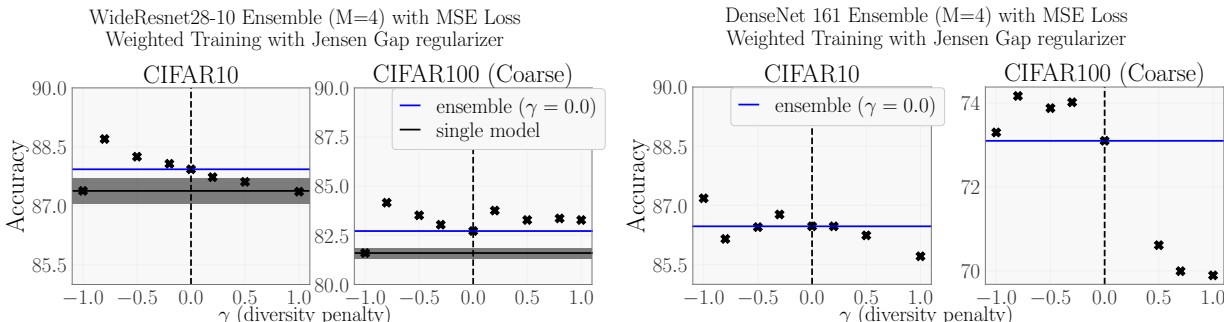

Figure 8: **Regardless of model capacity, encouraging predictive diversity does not hurt performance with MSE loss**. We also train deep regression ensembles using Eq. (2) with to MSE loss, on CIFAR10 and CIFAR100 (coarse labels). As predicted by our hypothesis, encouraging diversity does not hurt predictive performance in regression.

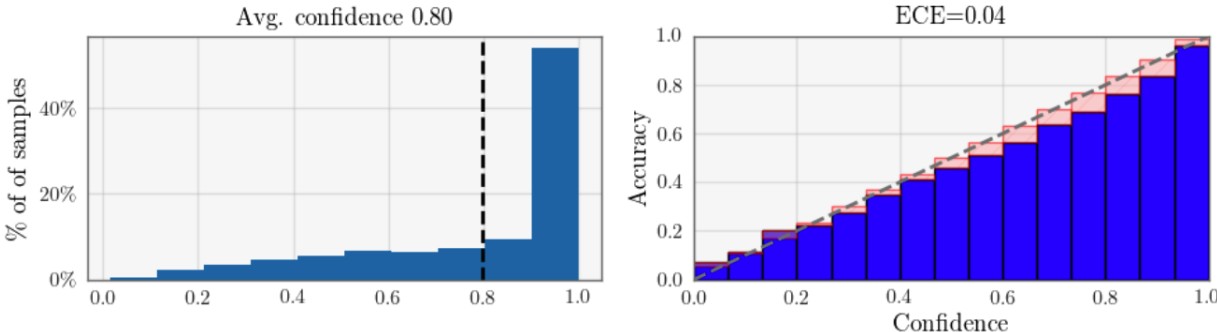

Figure 9: **Standard models make confident and accurate predictions on ImageNet**. An ensemble of ResNet 50 models on ImageNet makes the majority of its validation set predictions (∼55%) with a confidence between 0.9 and 1, ∼95% of which are correct predictions. These results indicate that even without reaching 0% training error, large capacity models are already susceptible to Observation 2.

contrast, the effect of discouraging predictive diversity appears to be dataset specific, leading to uniformly worse performance on CIFAR10 data, but leading to consistent improvements on CIFAR100, as in the case with large-model classification ensembles.

### D.3 ImageNet predictions

Although computational contraints prevent us from running extensive diversity regularization experiments on ImageNet, we provide preliminary evidence that we expect currently popular models to express the same trends beyond Table 1. In Fig. 9, we examined the predictions made by a ResNet 50 model on the ImageNet validation set, and we observe that it already makes a large proportion of its predictions with high confidence, including those that are highly accurate. These features of single model predictions would suggest that standard large models trained on very large datasets like ImageNet are most likely also susceptible to the pathologies we observe in Table 1.

### D.4 Label smoothing

In Fig. 2 we showed that high capacity models suffer when diversity is encouraged during training across all datapoints. We argued that encouraging diversity across all datapoints, and not only the errors, is particularly harmful to high capacity models which tend to produce high confidence predictions, whether the predictions are correct or not. Thus, the losses we incur when encouraging diversity in correct predictions outweigh any gains from encouraging diversity in incorrect predictions. A natural question that arises is encouraging

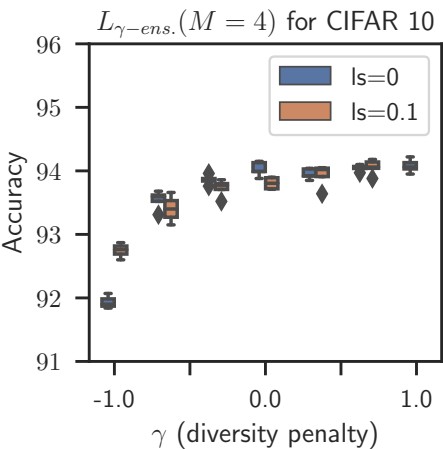

Figure 10: **Label smoothing mitigates impact of diversity encouragement, but maintains the same trend**. Treating our diversity regularized deep ensembles as a baseline ($ls = 0$, blue markers), we examine the impact of label smoothing $ls = 0.1$. Although higher levels of label smoothing mitigate the performance deficit, the overall trend of performance loss with predictive diversity remains.

diversity will be less harmful if the predictions are less confident, as was the case for low capacity models in Fig. 2. To test this hypothesis, we train ensembles using label smoothing (Müller et al., 2019), which softens the confidence of the predictions during neural network training. Fig. 10 shows that if we control for the confidence level via label smoothing while training ensembles, we do not see a significant change in performance in the results. Future work can explore whether stronger label smoothing or other techniques to temper the confidence of neural networks such as mixup (Zhang et al., 2017) or temperature scaling (Guo et al., 2017) can yield significant changes.

### D.5 CIFAR10 Out of distribution (OOD) data

Beyond InD test performance, deep ensembles are considered a guaranteed method to improve performance in particular under out of distribution (OOD) data, where the test data shifts significantly away from the training distribution (Gustafsson et al., 2020; Lakshminarayanan et al., 2017; Ovadia et al., 2019). We evaluated all of our CIFAR10 trained ensembles and individual models in Fig. 2 on CIFAR10.1, a shifted test dataset for models trained on CIFAR10 (Recht et al., 2018).

The results in Fig. 16 show the same conclusions we saw on InD data in Fig. 2: a stark drop in ensemble performance with a lower diversity penalty and a potential increase in performance with an increased penalty for $0 > \gamma > 1$, relative to $\gamma = 0$ unmodified training. This result supports the conclusions of Abe et al. (2022b), which suggests that ensemble performance OOD is very tightly coupled to performance InD.

# E  Jensen gap for common loss functions

In this section we provide explicit derivations for the Jensen Gap as a measure of predictive diversity under various losses $\ell$.

## E.1  The mean squared error Jensen gap

The Jensen gap for the mean squared error loss is a scaled version of the sample variance. The following is a derivation of that fact:

$$
\frac{1}{M}\sum_{i=1}^{M}\left(\boldsymbol{f}_i(\boldsymbol{x}) - y\right)^2 - \left(\frac{1}{M}\sum_{i=1}^{M}\boldsymbol{f}(\boldsymbol{x}) - y\right)^2
$$

$$
= \frac{1}{M}\sum_{i=1}^{M}\left[(\boldsymbol{f}_i(\boldsymbol{x}))^2 - 2(\boldsymbol{f}_i(\boldsymbol{x}))y + y^2\right] - \left[(\bar{\boldsymbol{f}}(\boldsymbol{x}))^2 - 2(\bar{\boldsymbol{f}}(\boldsymbol{x}))y + y^2\right]
$$

$$
= \frac{1}{M}\sum_{i=1}^{M}\left[(\boldsymbol{f}_i(\boldsymbol{x}))^2 - 2(\boldsymbol{f}_i(\boldsymbol{x}))y\right] - \left[(\bar{\boldsymbol{f}}(\boldsymbol{x}))^2 - 2(\bar{\boldsymbol{f}}(\boldsymbol{x}))y\right]
$$

$$
= \frac{1}{M}\sum_{i=1}^{M}(\boldsymbol{f}_i(\boldsymbol{x}))^2 - \frac{1}{M}\sum_{i=1}^{M}2(\boldsymbol{f}_i(\boldsymbol{x}))y - (\bar{\boldsymbol{f}}(\boldsymbol{x}))^2 + 2(\bar{\boldsymbol{f}}(\boldsymbol{x}))y
$$

$$
= \frac{1}{M}\sum_{i=1}^{M}\left[(\boldsymbol{f}_i(\boldsymbol{x}))^2 - (\bar{\boldsymbol{f}}(\boldsymbol{x}))^2\right]
$$

$$
= \frac{M-1}{M}\left[\frac{1}{M-1}\sum_{i=1}^{M}\left[(\boldsymbol{f}_i(\boldsymbol{x}))^2 - (\bar{\boldsymbol{f}}(\boldsymbol{x}))^2\right]\right]
$$

$$
= \frac{M-1}{M}\,\overline{\mathrm{Var}}\left[\boldsymbol{f}_i(\boldsymbol{x})(\boldsymbol{x})\right],
$$

where $\bar{\boldsymbol{f}}(\boldsymbol{x}) = \frac{1}{M}\sum_{i=1}^{M}\boldsymbol{f}_i(\boldsymbol{x})$ is the sample mean and $\overline{\mathrm{Var}}$ is the sample variance.

## E.2  Decomposition of cross entropy Jensen gap

Throughout the paper, we use the Jensen gap for cross entropy loss as a metric of diversity on test data, as well as a regularizer for predictive diversity. Here we show that the Jensen gap is an information theoretic quantification of diversity. The Jensen gap is given by:

$$
\frac{1}{M}\sum_{i=1}^{M}\left[\mathrm{CE}(\boldsymbol{f}_i(\boldsymbol{x}),\ y)\right] - \left[\mathrm{CE}(\bar{\boldsymbol{f}}(\boldsymbol{x}),\ y)\right]
$$

$$
= \frac{1}{M}\sum_{i=1}^{M}\left[-\log \boldsymbol{f}_i(\boldsymbol{x})^{(y)}\right] - \left[-\log \bar{\boldsymbol{f}}(\boldsymbol{x})^{(y)}\right]
$$

$$
= \frac{1}{M}\sum_{i=1}^{M}\left[-\log \boldsymbol{f}_i(\boldsymbol{x})^{(y)}\right] + \log\left[\frac{1}{M}\right] + \log\left[\sum_{j=1}^{M}\boldsymbol{f}_j(\boldsymbol{x})^{(y)}\right]
$$

$$
= \sum_{i=1}^{M}\frac{1}{M}\left[\log\frac{1}{M} - \log\left[\frac{\boldsymbol{f}_i(\boldsymbol{x})^{(y)}}{\sum_{j=1}^{M}\boldsymbol{f}_j(\boldsymbol{x})^{(y)}}\right]\right] \tag{9}
$$

Eq. (9) can be interpreted as the KL divergence between two categorical distributions. The first distribution represents the probability of sampling an ensemble member uniformly at random $(1/M)$. The second distribution represents the probability of sampling an ensemble member proportional to its correct class prediction. Eq. (9) will be minimized when these two distributions are equal, which will only happen if all the ensemble members predict the correct class equally. Conversely, Eq. (9) will be larger when the component model predictions differ from one another.

Fig. 11 illustrates the high degree of correlation between the Jensen Gap diversity in Eq. (3) and other diversity metrics used in the literature, including the KL divergence (Lee and Seung, 2000), pair-wise correlation

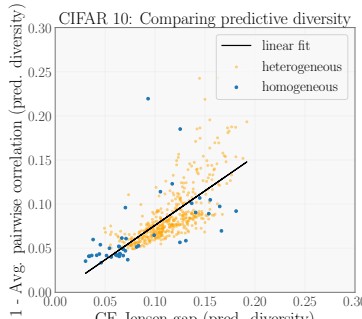 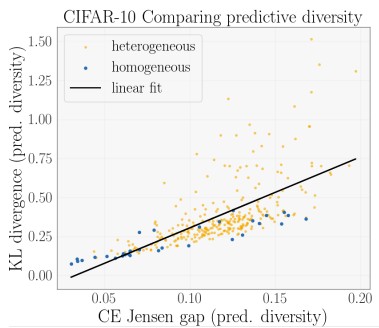 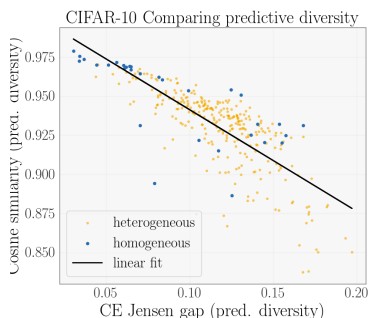

Figure 11: **CE Jensen gap is correlated with standard measures of diversity**. We compare the Jensen gap notion of predictive diversity (here with cross entropy loss) to other standard metrics of predictive diversity on the homogeneous and heterogeneous CIFAR10 ensembles shown in Fig. 6. We show that CE Jensen gap predictive diversity is highly correlated with diversity metrics based on the average pairwise correlation between the predictions of ensemble members (**left**), diversity based on KL divergence (**middle**), and diversity based on cosine similarity (**right**)- linear fits are given by black lines.

| Diversity Encouragement... | Hurts | Neutral | Helps |
|---|---|---|---|
| Small NN Classification Ensembles | 12/85 (14%) | 7/85 (8%) | 66/85 (78%) |
| Large overconfident NN Classification Ensembles | 82/134 (61%) | 38/134 (29%) | 14/134 (10%) |

Table 3: **Large capacity deep ensembles respond differently to diversity encouragement**. Across a large number of experimental settings, we see that encouraging predictive diversity has opposite effects on large capacity deep ensembles vs. other types of ensembles. The classification of ensembles listed is calculated relative to 2 standard error of standard ensemble performance. Note these results do not include MLPs.

| Diversity Discouragement... | Hurts | Neutral | Helps |
|---|---|---|---|
| Small NN Classification Ensembles | 46/82 (56%) | 33/82 (40%) | 3/82 (4%) |
| Large Overconfident NN Classification Ensembles | 40/155 (26%) | 78/155 (50%) | 37/155 (24%) |

Table 4: **Large capacity deep ensembles respond differently to diversity discouragement**. Across a large number of experimental settings, we see that discouraging predictive diversity has opposite effects on large capacity deep ensembles vs. other types of ensembles. The classification of ensembles listed is calculated relative to 2 standard error of standard ensemble performance. Note these results do not include MLPs.

(Kuncheva and Whitaker, 2003), and cosine similarity (Djolonga et al., 2021). The models in Fig. 11 include the same CIFAR10 models in Fig. 6.

# F   Additional analysis of ensembles trained with diversity mechanisms

In this section, we provide additional replication results for combinations of model architectures, datasets, and regularizers not shown in the main text, as well as evaluation metrics beyond test accuracy. Tables 3 and 4 summarize our findings. Scripts to generate results in the style of Fig. 2 (as well as that figure itself) are of the form `ensemble_attention/scripts/vis_scripts/all_weights_{classifier_name}_{dataset_name}.py`.

| Diversity Encouragement... | Hurts | Neutral | Helps |
|---|---|---|---|
| Small NN (Jensen Gap) | 0/23 (0%) | 6/23 (26%) | 17/23 (74%) |
| Small NN (Variance) | 0/20 (0%) | 0/20 (0%) | 20/20 (100%) |
| Small NN (JSD 1 vs. All) | 11/23 (48%) | 2/23 (9%) | 10/23 (43%) |
| Small NN (JSD Uniform) | 1/20 (5%) | 3/20 (15%) | 16/20 (80%) |
| Large NN (Jensen Gap) | 23/34 (68%) | 8/34 (24%) | 3/34 (8%) |
| Large NN (Variance) | 25/35 (72%) | 5/35 (14%) | 5/35 (14%) |
| Large NN (JSD 1 vs. All) | 18/31 (58%) | 12/31 (39%) | 1/31 (3%) |
| Large NN (JSD Uniform) | 17/34 (50%) | 10/34 (29%) | 7/34 (21%) |

Table 5: **Performance per regularizer (encourage)**.

### F.1 Analyzing differences between diversity mechanisms

Although the diversity mechanisms discussed in Sec. 4 all demonstrate broadly similar behavior across the models that we study, there are some notable differences between them that may be useful for further study. We discuss some of these differences here.

**Effective ranges of regularizer strengths.** To further study the behavior of these regularizers it would be useful to consider the differences in the effective ranges of regularization strength. In Jeffares et al. (2023), the authors show that the Jensen gap loss demonstrates pathological behavior for values of $\gamma < -1$ when derived from any loss function, not only the cross entropy and squared error loss that we consider here. Likewise, Mishtal and Arel (2012) show analytically that for the JSD 1 vs. All regularizer, we can only ensure the existence of a minimum of the loss function for $\gamma > -4$ with the square error loss. They provide experimental evidence that this bound applies to the cross entropy loss as well. We are not aware of previously reported bounds for the Variance regularizer or the JSD Uniform regularizer, and we did not run into stability issues across the range of regularization strengths that we tested with these mechanisms. Developing such bounds for all regularizers (and diversity discouragement) would be useful to understand how make more detailed comparisons between individual regularizers which would respect these differences in the effective ranges of their regularization strengths.

**Impact on network predictions.** These diversity mechanisms also differ in what aspect of the prediction that they seek to diversify. In particular, both Jensen gap and Variance regularizers specifically seek to diversify the probability assigned to the correct prediction, $f_i^y(\boldsymbol{x})$ during training, whereas the others seek to diversify the entire vector of predictions, $f_i(\boldsymbol{x})$. While we observed no significant differences between these approaches in the experiments presented within this paper, this is an important distinction that merits study in future work.

**Empirical effectiveness of individual mechanisms.** Finally, we break down the results presented in Tables 3 and 4 into results for each individual mechanism, presented in Tables 5 and 6. Although there is some variation in the effectiveness (or lack thereof) of individual diversity mechanisms in impacting the performance of ensembles, our main conclusions hold in all cases.

### F.2 Decomposing performance of ensembles trained with diversity mechanisms (additional results)

Fig. 12 demonstrates that the trends described in Fig. 3 also apply on the CIFAR100 dataset.

Although not a mathematical decomposition of ensemble performance, in Fig. 13 we also analyze the relationship between predictive diversity (measured as CE Jensen gap) and average single model accuracy, for

| Diversity Discouragement... | Hurts | Neutral | Helps |
|---|---|---|---|
| Small NN (Jensen Gap) | 11/19 (58%) | 7/19 (37%) | 1/19 (5%) |
| Small NN (Variance) | 19/20 (95%) | 1/20 (5%) | 0/20 (0%) |
| Small NN (JSD 1 vs. All) | 14/23 (61%) | 8/23 (35%) | 1/23 (4%) |
| Small NN (JSD Uniform) | 6/20 (30%) | 12/20 (60%) | 2/20 (10%) |
| Large NN (Jensen Gap) | 13/50 (26%) | 23/50 (46%) | 11/50 (22%) |
| Large NN (Variance) | 13/37 (35%) | 17/37 (46%) | 7/37 (19%) |
| Large NN (JSD 1 vs. All) | 5/31 (16%) | 17/31 (55%) | 9/31 (29%) |
| Large NN (JSD Uniform) | 9/38 (24%) | 18/38 (47%) | 11/38 (29%) |

Table 6: **Performance per regularizer (discourage)**.

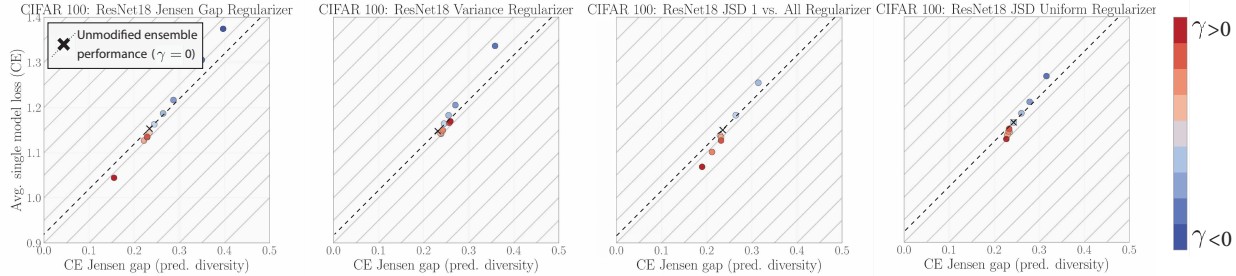

Figure 12: **Measuring predictive diversity for diversity regularized ensembles on CIFAR100**. We apply diversity regularized deep ensemble training to large capacity ResNet 18 models on CIFAR100 as well. Same conventions and conclusions as in Fig. 3.

the same models shown in Fig. 3. We see that ensemble member performance measured via average ensemble member accuracy closely reflects the decomposition presented in Fig. 3.

### F.3 Figs. 2 and 3 (Diversity regularization for additional large-capacity models)

We show that the results in Fig. 2 hold for additional large capacity architectures (DenseNet 121, VGG 11, WideResNet 18, VGG 13) for both CIFAR10 and CIFAR100 in Figs. 14 and 15, and OOD data (Fig. 16).

### F.4 Fig. 2 (Diversity regularization for additional small-capacity models)

We show that the results for low capacity models in Fig. 2 hold for additional architectures trained on CIFAR10 and CIFAR100 in Fig. 18 and Fig. 19.

### F.5 Calibrated performance metrics

We show that the results in Fig. 2 generalize to calibrated performance metrics: NLL (Fig. 20), Brier Score (Fig. 21), and ECE (Fig. 22).

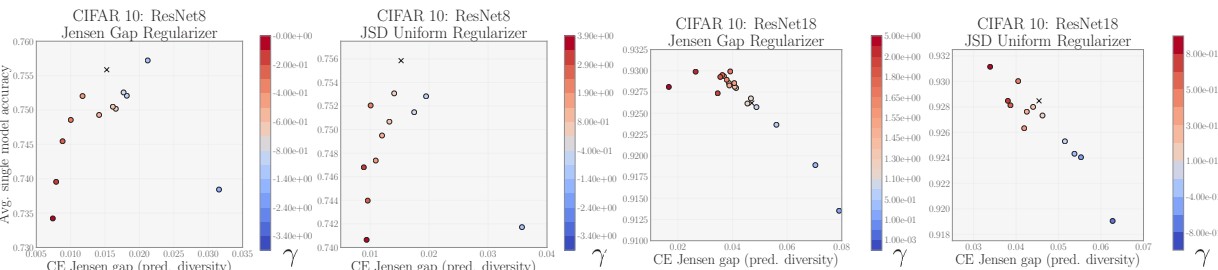

Figure 13: **Measuring predictive diversity and component model accuracy in diversity regularized ensembles.** As in Fig. 3, each marker represents a ResNet 8 (**left panels**) or ResNet 18 (**right panels**) ensemble trained with a diversity intervention on CIFAR10, with colors representing $\gamma$ values, and the $\times$ representing standard ensemble performance. We omit diagonal level set lines, as there is no decomposition of ensemble performance into Jensen gap predictive diversity and accuracy. Nevertheless, we similar patterns to Fig. 3: for small-network ensembles (**left**), diversity encouragement successfully creates more diverse ensembles, which can sometimes lead to ensemble members with higher accuracy as well, while diversity discouragement leads to a loss in performance. Meanwhile, for large-network ensembles (**right**), diversity encouragement leads to a decline in performance, but diversity discouragement appears to generate notably better ensemble members across the range of tested regularization strengths.

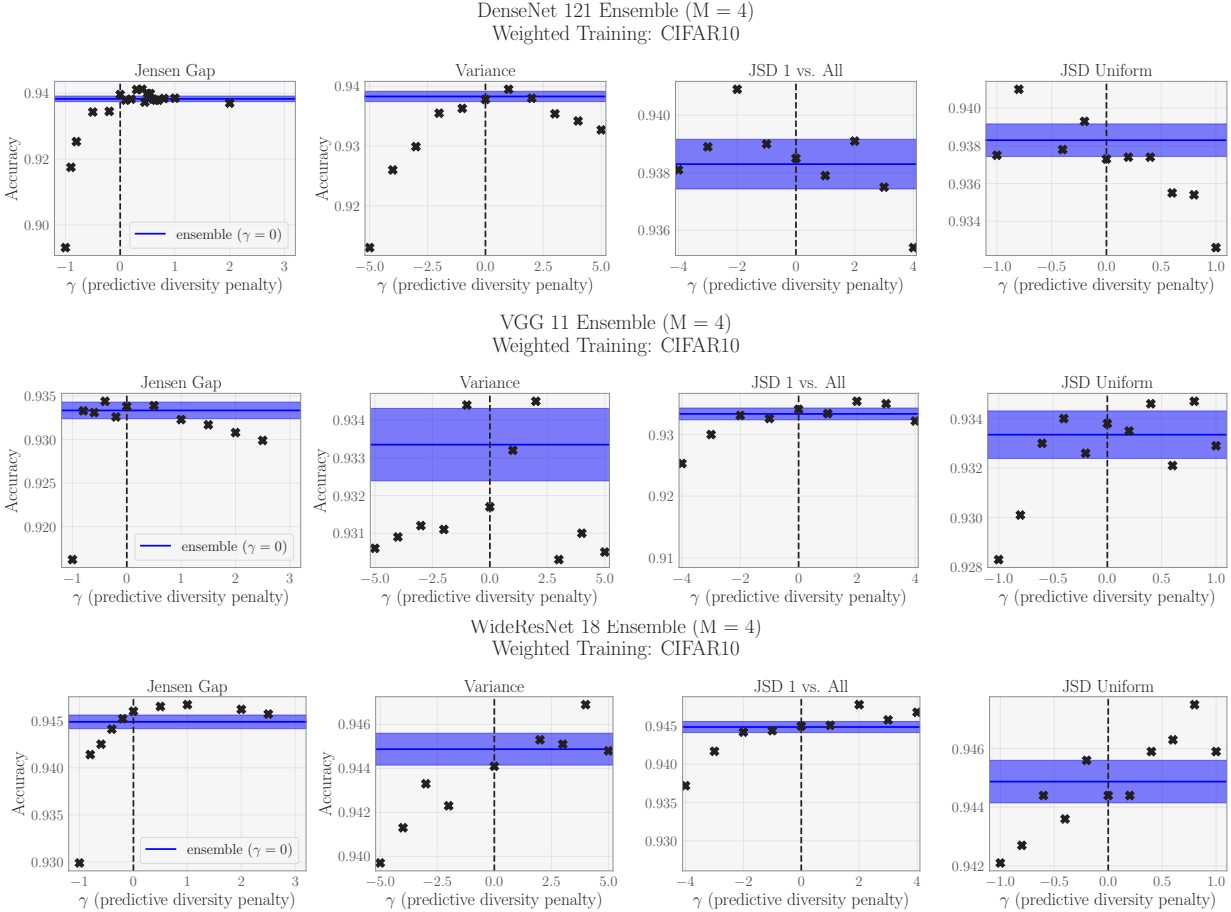

Figure 14: **Diversity encouragement hurts deep ensemble accuracy of large capacity models: additional architectures for CIFAR10**. We apply diversity regularized deep ensemble training to 3 other architectures on CIFAR10. Rows depict training for Densenet 121, VGG 11, and WideResNet 18. Same conventions and conclusions as Fig. 2.

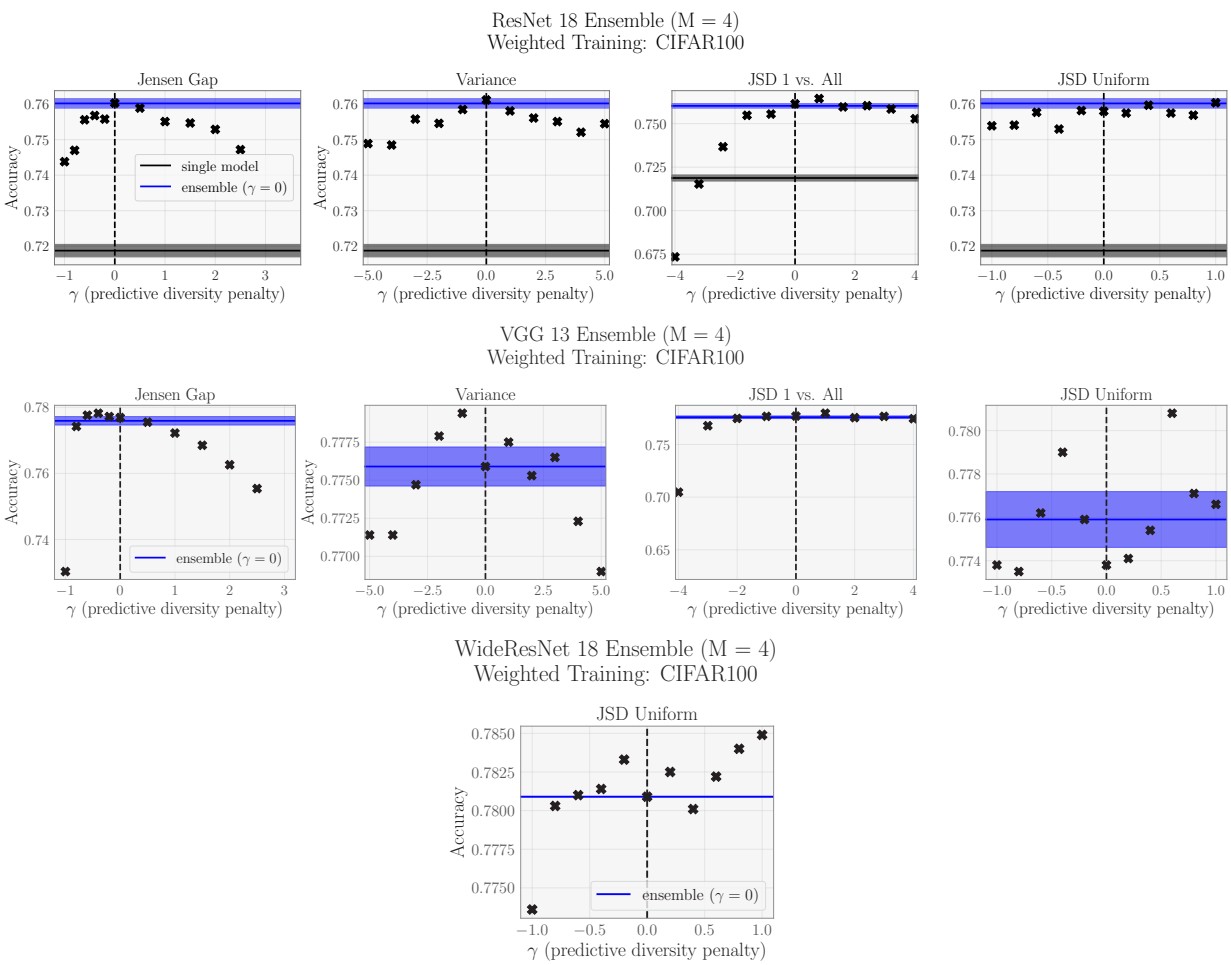

Figure 15: **Diversity encouragement hurts deep ensemble accuracy of large capacity models: CIFAR100**. We apply diversity regularized deep ensemble training to 2 other architectures on CIFAR100. Rows depict training for VGG 11 with all regularizers, and WideResNet 18 with the JSD Uniform regularizer. Same conventions and conclusions as Fig. 2.

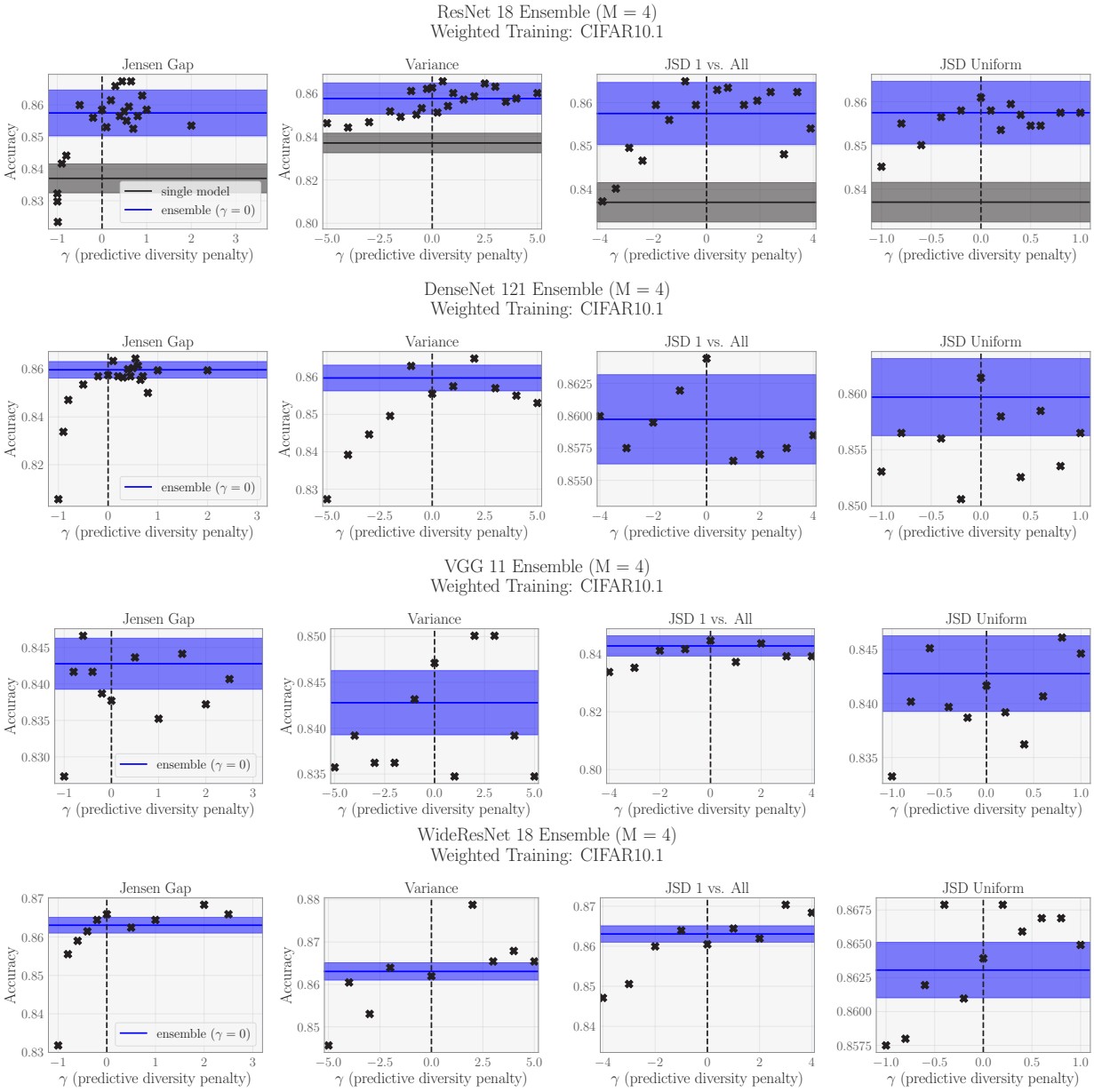

Figure 16: **Diversity regularization demonstrates the same trends on OOD data**. We evaluate large scale diversity regularized models trained on CIFAR10 (identical to Fig. 2) on an OOD test dataset, CIFAR10.1. Predictive performance follows the same trends demonstrated on the standard CIFAR10 test set.

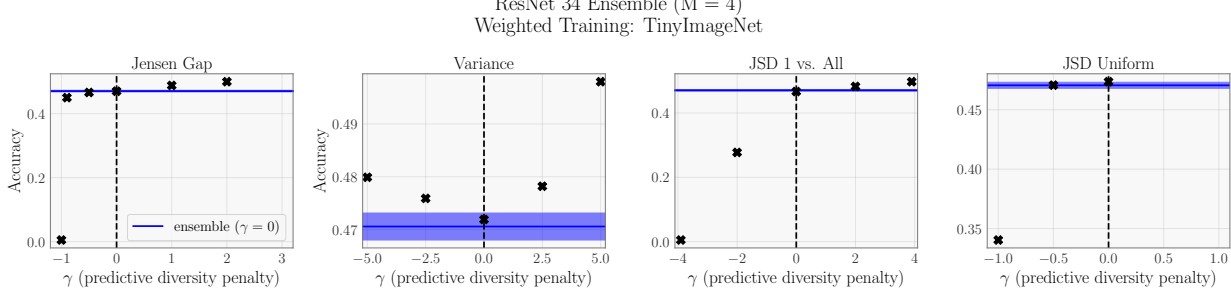

Figure 17: **Diversity encouragement hurts deep ensemble accuracy of large capacity models: TinyImageNet**. We apply diversity regularized deep ensemble training to another architecture on TinyImageNet. Panels depict training for ResNet 34 with all regularizers. Same conventions and conclusions as Fig. 2.

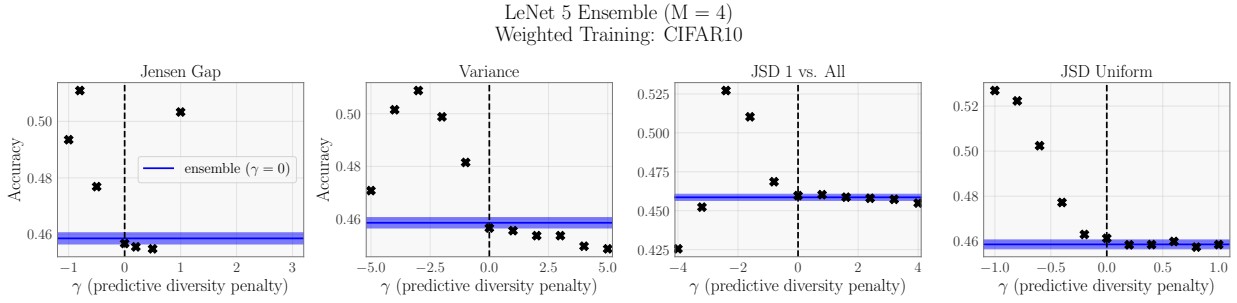

Figure 18: **Diversity encouragement improves low capacity models: additional architecture for CIFAR10.** We apply diversity regularized deep ensemble training to LeNet 5 on CIFAR10, and measure predictive accuracy. Same conventions and conclusions as in Fig. 2.

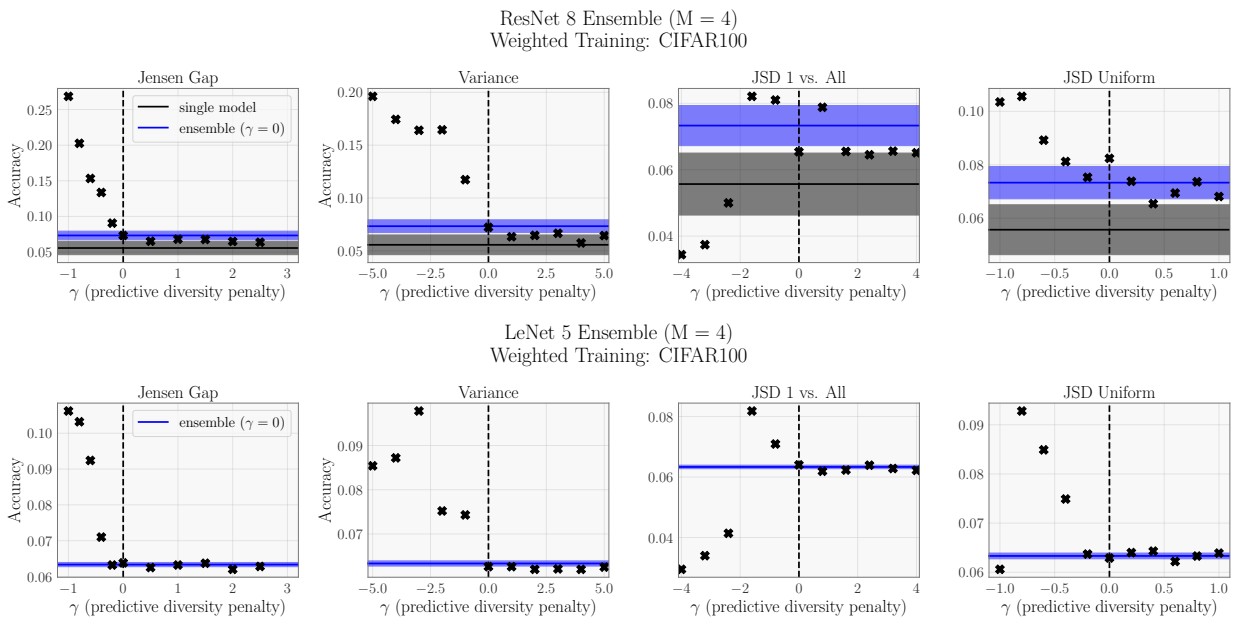

Figure 19: **Diversity encouragement improves low capacity models: ResNet 8 and additional architecture for CIFAR100.** Same conventions and conclusions as Fig. 2.

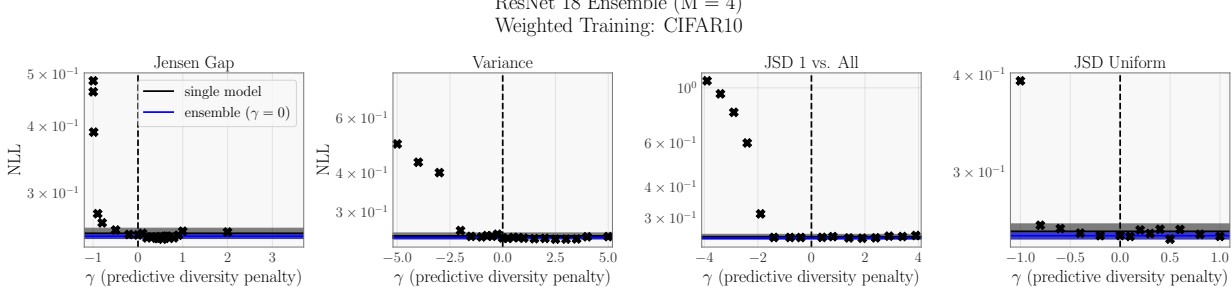

Figure 20: **Diversity encouragement hurts deep ensemble NLL of large capacity models**. We apply diversity regularized deep ensemble training to ResNet 18 on CIFAR10, and measure NLL. Same conventions and conclusions as in Fig. 2.

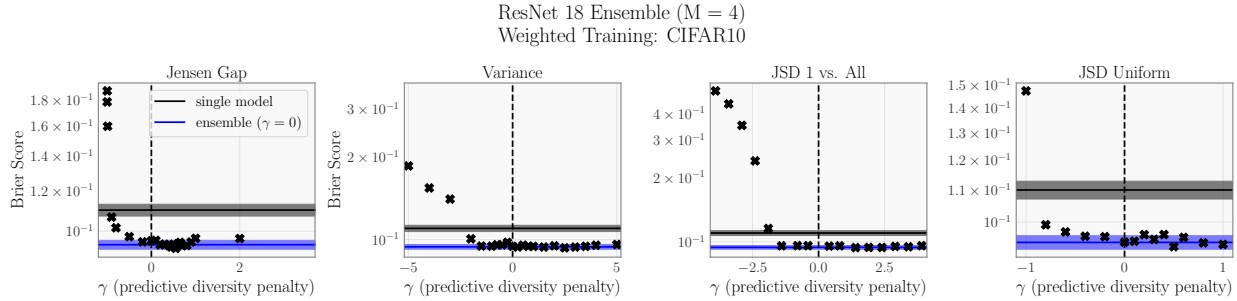

Figure 21: **Diversity encouragement hurts deep ensemble Brier score of large capacity models of large capacity models**. We apply diversity regularized deep ensemble training to ResNet 18 on CIFAR10, and measure Brier score. Same conventions and conclusions as in Fig. 2.

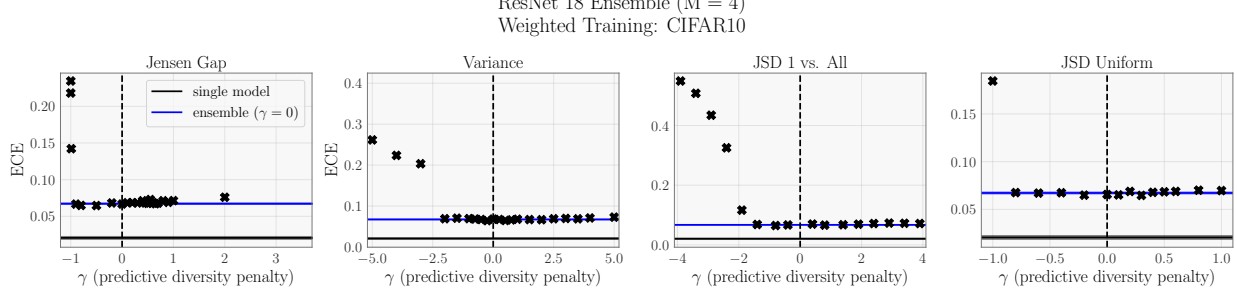

Figure 22: **Diversity encouragement hurts deep ensemble ECE of large capacity models**. We apply diversity regularized deep ensemble training to ResNet 18 on CIFAR10, and measure expected calibration error. Same conventions and conclusions as in Fig. 2.

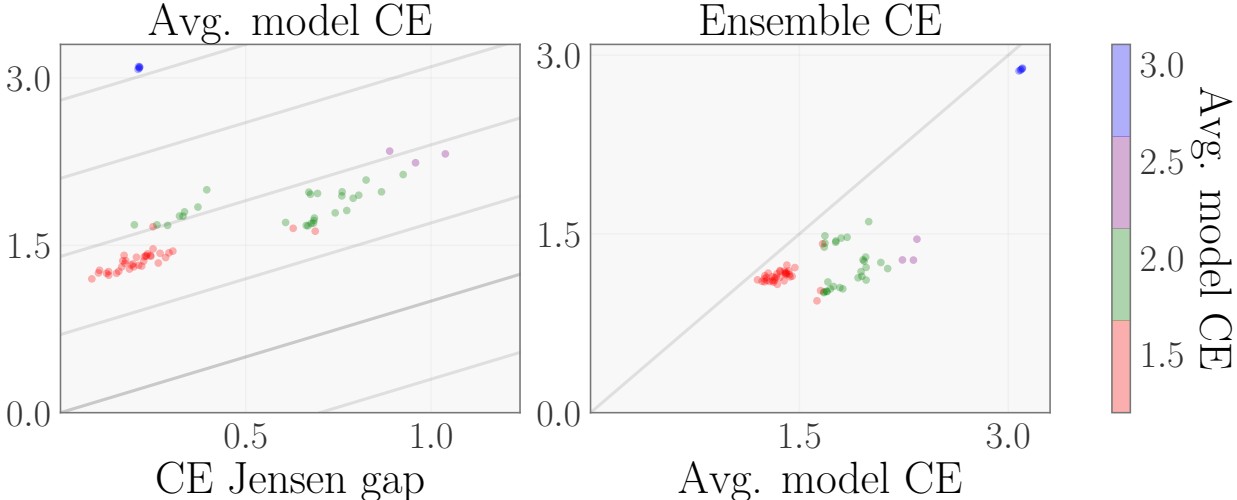

Figure 23: **The best ensemble performance is obtained by ensembles with the best models**. Same models as Fig. 5 where the models are evaluated on ImageNet V2MF. **Left:** Diversity vs. average single model performance decomposition, as in Fig. 1. **Right:** Average single model vs. ensemble performance decomposition as in Fig. 5. For each ensemble color indicates average CE of component models, as given in colorbar.

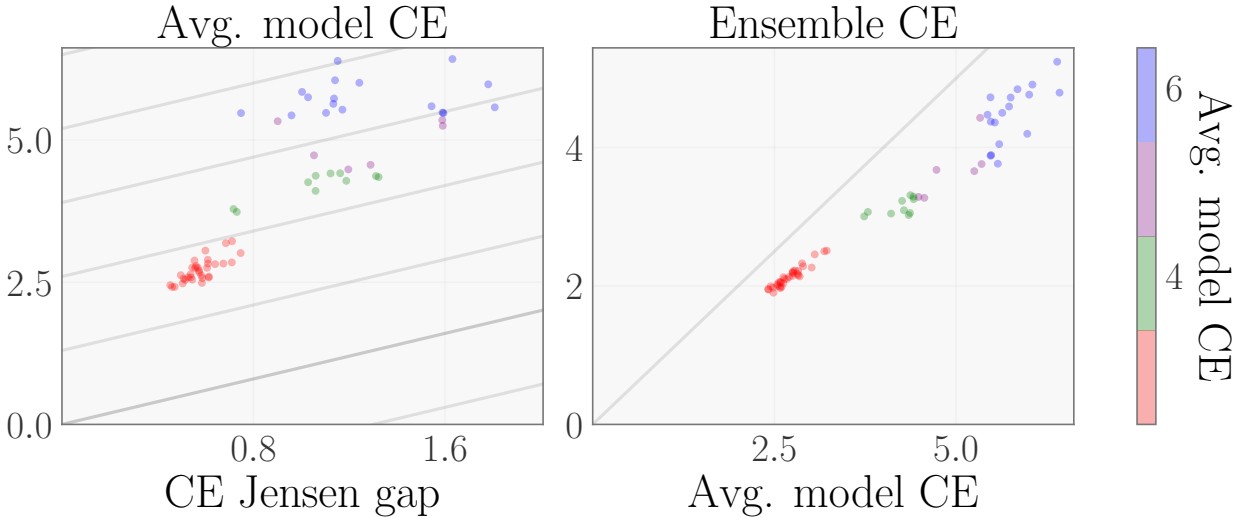

Figure 24: **The best ensemble performance is obtained by the best models**. Same models as Fig. 5 where the models are evaluated on ImageNet C gaussian-noise-3. Conventions as in Fig. 23.

## G  The best ensembles have low predictive diversity across a variety of test datasets

Fig. 5 illustrates the tradeoff between average single model performance and Jensen gap for multiple models trained independently. In this section we extend the results from Fig. 5 and include the comparison between average single model and the ensemble loss for a variety of OOD datasets, including ImageNetV2 (Recht et al., 2019) and additional ImageNet-C datasets (Hendrycks and Dietterich, 2019). Overall Figs. 23 to 25 show that forming better performing ensembles from better component models comes at the cost of predictive diversity.

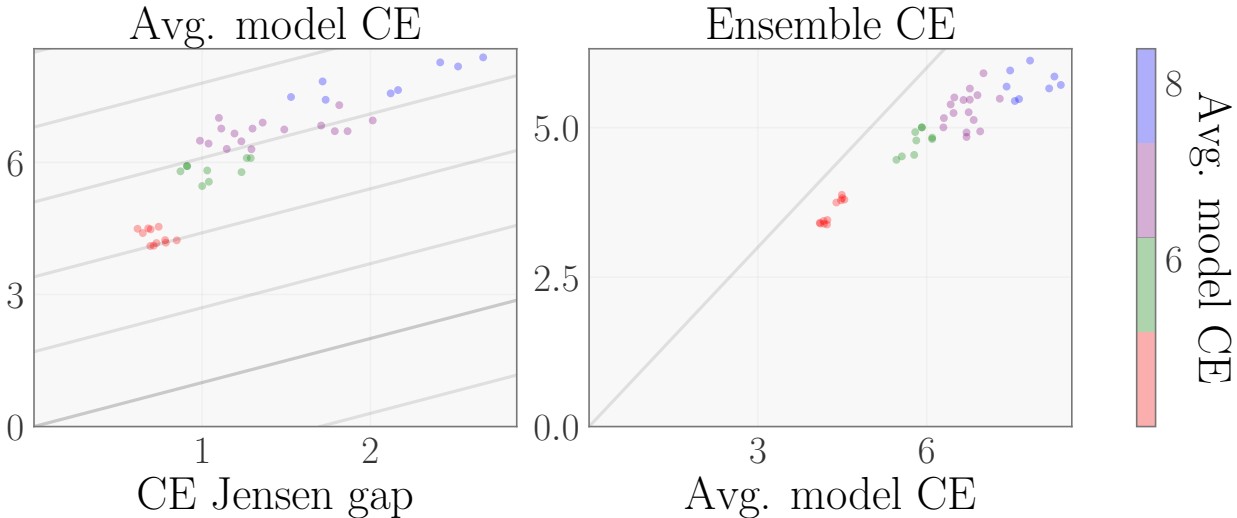

Figure 25: **The best ensemble performance is obtained by the best models**. Same models as Fig. 5 where the models are evaluated on ImageNet C gaussian-noise-5. Conventions as in Fig. 23

## H   Toy experiments

How should we expect error decorrelation to impact diversity and single model performance for accurate, confident predictions? We provide additional intuition for this question with three known methods of improving ensemble diversity in Fig. 26, where we consider the effect of each of three diversity interventions on the quality of a single prediction. We decompose ensemble performance on this single prediction as shown in Fig. 1. Each diversity intervention is indexed with a scale parameter $s$, which controlled the level of predictive diversity expressed. In all cases, we considered the diversity between three ensemble member predictions $c_i$, in a three class classification task, and we assume for simplicity that the correct class is 0, implying that the perfect prediction is $e_0$, where $e_i$ is the $i$th member of the standard basis in $\mathbb{R}^3$.

**Geometric diversity.** As a limiting case, we considered how smoothly interpolating from a perfect prediction to one with maximal diversity. In this case, our three predictions $c_i$ are:

$$c_i = (1 - s) * (e_0) + s * (e_i) \tag{10}$$

**Logit diversity.** Next, we considered how adding predictive diversity in logit space could affect the predictions in probability space. In this case, our three predictions are identically defined as:

$$c_i = \text{softmax}(10 * e_0 + \epsilon); \epsilon \sim \mathcal{N}(0, s) \tag{11}$$

**Dirichlet diversity.** Finally, we consider sampling predictions from a dirichlet distribution, directly in the simplex:

$$c_i \sim \text{Dir}(\alpha); \alpha = \frac{1}{10} * [s, (1 - s)/2, (1 - s)/2] \tag{12}$$

For both of our sampling based simulations, we calculate CE Jensen Gap and average single model NLL across 100 iterates, and plot the averages.

We see that regardless of our choice of diversity mechanism, average single model loss degrades more quickly than diversity can increase, providing experimental verification of our intuition that diversity can only hurt performance at extreme points of the simplex. We provide code to replicate this figure at `interp_ensembles/scripts/visualize_diversity/figure_gen.py`.

# I Counterfactual accuracy analysis

For both ensembles of trees and deep ensembles with diversity interventions, we consider the *counterfactual accuracy*: the accuracy of the ensemble in the absence of the intervention, as a function of the predictive diversity achieved by the intervention. Our diversity measure in all cases is the cross entropy Jensen Gap, (Appx. E) applied to individual datapoints in the test set. More specifically, for each datapoint in the test set, we calculate:

- the Jensen gap predictive diversity expressed by ensemble members on this test point.

- the average accuracy across ensemble members in the absence of diversity interventions (the counterfactual accuracy).

The top panel of each figure depicts a histogram of the predictive diversity between ensemble members, smoothed with a gaussian kernel. To generate the bottom panel, we use logistic regression models to predict counterfactual accuracy as a function of predictive diversity. Code to replicate this analysis given a set of test set logits is given in `ensemble_attention/notebooks/counterfactual_accuracy.ipynb`.

## I.1 Trees

For ensembles of trees, we consider a single tree as the "base ensemble", as there is little stochasticity in the training of trees without specific diversity interventions. We consider trees of depth 1 to 12, fit to the MNIST dataset. As component trees become deeper, ensembles express more predictive diversity. To avoid binary predictions (for which our diversity metric is undefined), we pad all predictions with $\epsilon = 1^{-10}$ and renormalize.

## I.2 Deep ensembles

For deep ensembles, we consider a standard deep ensemble (formed over initialization) as the base ensemble. As diversity interventions, we consider each of the diversity interventions described in Sec. 4 with values of $\gamma < 0$. Note that $\gamma = 0$ corresponds to a standard deep ensemble. As we consider more negative values of $\gamma$, ensembles express more predictive diversity.

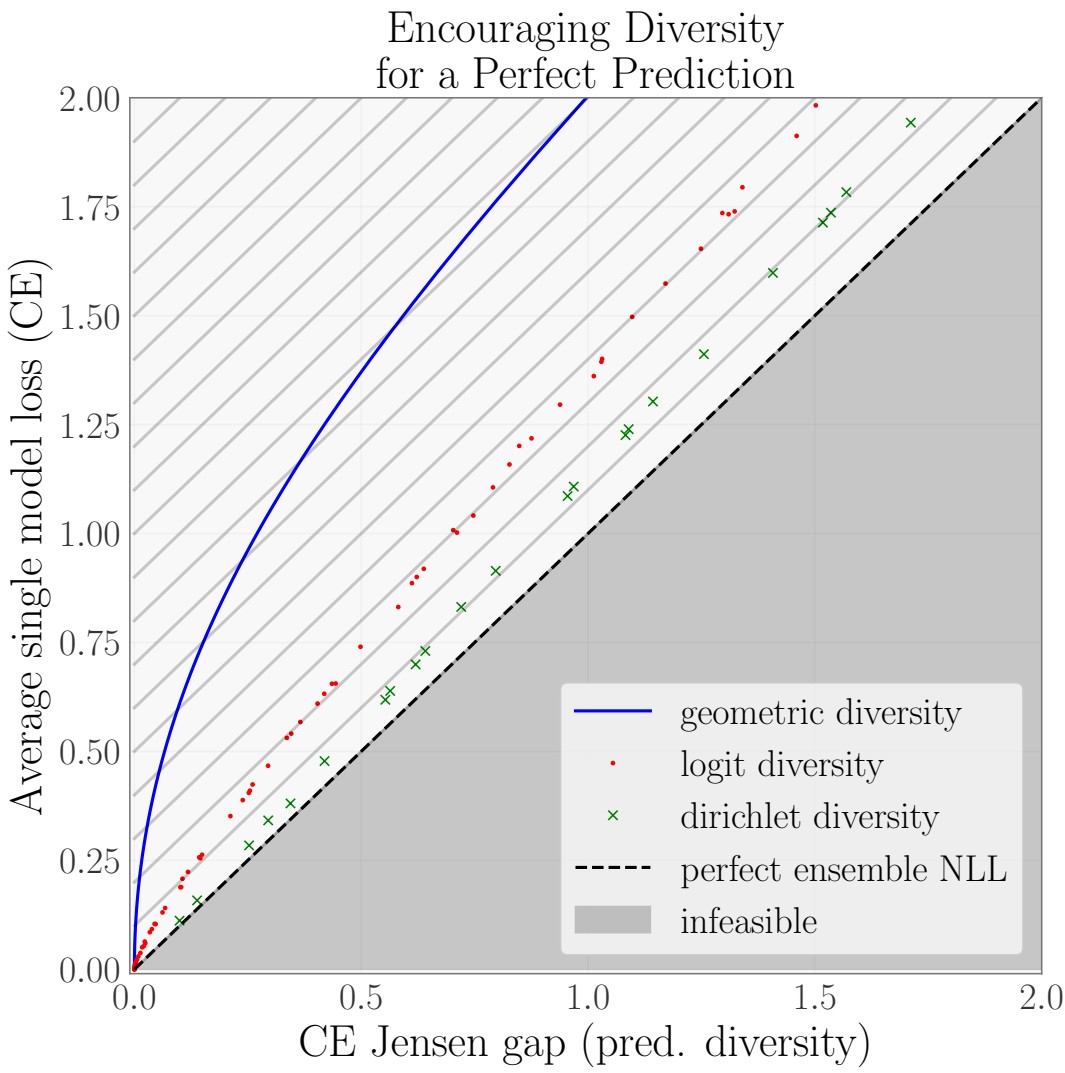

Figure 26: **Regardless of encouragement strategy, diversity is impossible at extreme points.** Under 3 different methods of adding diversity to a perfect ensemble prediction (geometric scaling, sampling in logit space, or sampling from a dirichlet distribution), *any* of these strategies will degrade average single model loss more rapidly than increases in predictive diversity, leading to worse ensemble performance overall.

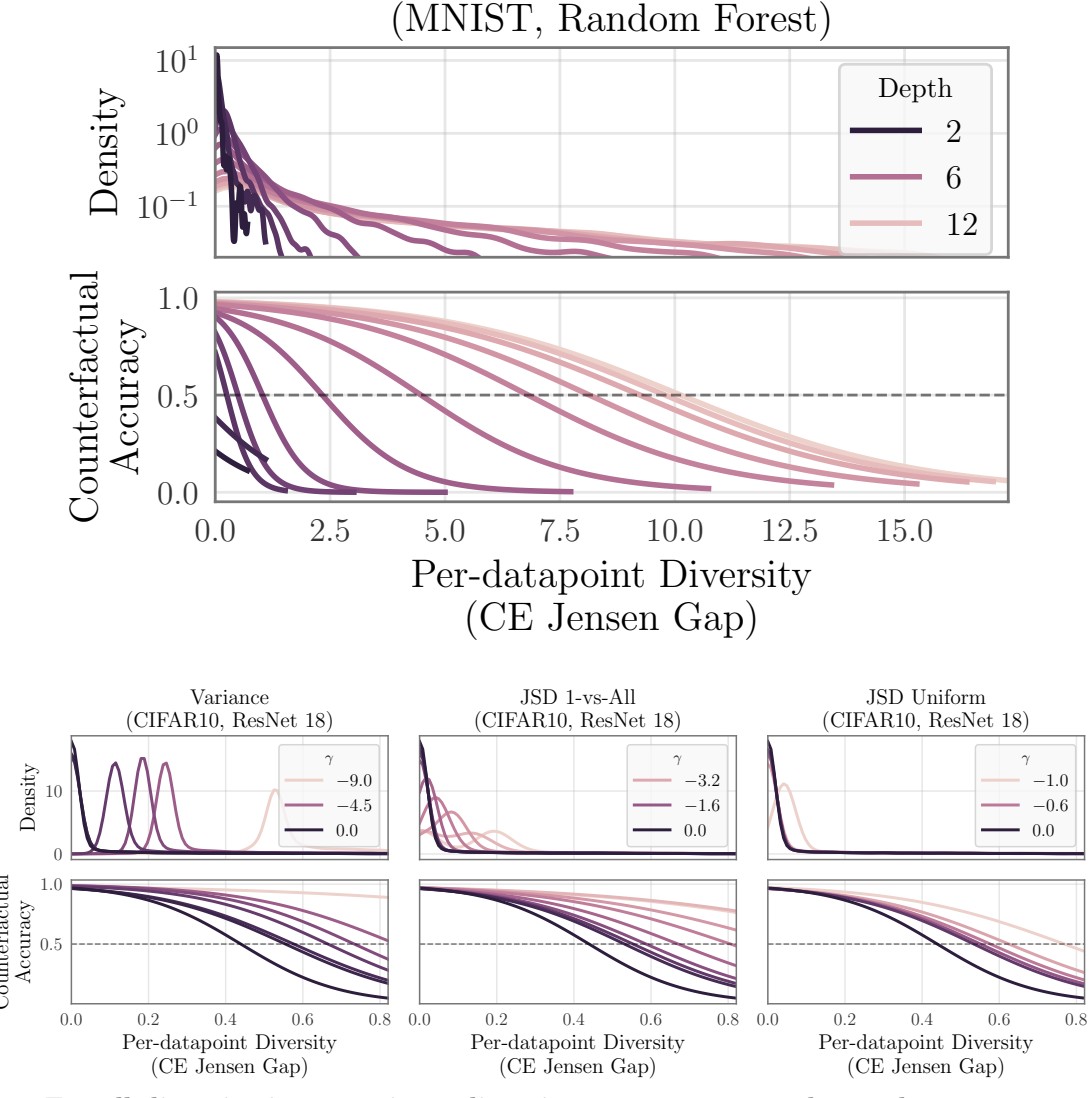

Figure 27: **For all diversity interventions, diversity encouragement decorrelates correct predictions.** Just as with Fig. 4, we find that methods which encourage a greater level of predictive diversity decorrelate correct ensemble predictions, where we present results here for random forests (**top**), as well as the three diversity regularizers not shown in the main text applied to ResNet 18 (**bottom**).

