# OpenReview forum: "Pathologies of Predictive Diversity in Deep Ensembles"
_TMLR — Accepted by TMLR_

### Review · Reviewer_2UB5 · 2023-10-19

**Summary Of Contributions:**

The paper examines the phenomenon of trading off between diversity and accuracy for ensembles of deep learning models. Experiments come to a conclusion, that encouraging diversity for strong models lead to the degradation of accuracy in ensembles. The conclusion is supported by numerous experiments including experiments for TinyImageNet and ImageNet

**Audience:**

Yes

**Broader Impact Concerns:**

-

**Claims And Evidence:**

Yes

**Requested Changes:**

- Make an ablation study on the training procedure, look at the relevant procedure for training
- Provide accuracy values as well as CE for later experiments

**Strengths And Weaknesses:**

Strengths:
- The authors have written a nice and easy to follow paper with numerours experiments provided in the appendix and in the main text
- The number of experiments is high, and they approach the problem from different angles, leading to the same conclusion

Weaknesses:
- I would agree with the main conclusion of the paper and with the statement that it is supported with experiments, however, we see, that usage of the variance as the diversity measure leads to better models (Figures 13-14). It means, that even larger models aren't large enough to completely ignore the diversity issues.
- Reporting only CE loss can limit the explanatory power of the experiments, as one-hot true label is an incorrect categorical distribution for an example. In my opinion, reporting of Accuracy lead to deeper understanding and lead to better insights in what is going on (e.g. provided in Figures 12-20 vs Figures 21-23)
- The training procedure can harm the conclusions: simultaneous training of models in an ensemble often leads to the performance degradation on average. The models should be trained indendently with little intersection over losses for them (e.g. boosting-style training).
- Figure 17 y-axis label is wrong

---

> ### Author Response · Authors · 2023-11-16
> **Thank you for your feedback!**
>
> Thank you for your review of our paper! We appreciate your feedback and we hope we are able to address it in our responses below.
>
> > I would agree with the main conclusion of the paper and with the statement that it is supported with experiments, however, we see, that usage of the variance as the diversity measure leads to better models (Figures 13-14). It means, that even larger models aren't large enough to completely ignore the diversity issues.
>
> Importantly, we would like to point out that our claims are not specific to individual models or regularizers, and rather that the goal of our work is to summarize that the broad trend of diversity encouragement techniques appears ineffective/detrimental to modern deep learning architectures. However, we also observe in additional experiments on MLPs with tabular data that (please see response to reviewer 5PH2) that as we encourage diversity on larger sized MLPs, accuracy can first reach a local maximum before decreasing as a function of regularization strength (see `kl_weight_forest_mlp_big.pdf` in supplementary material). This local optimum goes away when we consider even larger models (se `kl_weight_forest_mlp_bigger.pdf`). It could indeed be the case the some models paired with the Variance regularizer are not large enough to observe uniformly worse performance under diversity regularization. We will discuss these observations in an updated revisions, but also note that given the trend towards using larger and more powerful component models, we expect that the impact of diversity encouragement will only become more harmful in the future.
>
> We will also include an analysis of performance per diversity regularizer in the appendix, as suggested by reviewer Dz3X, where we can better contrast individual regularizers against one another. From this breakdown, (in particular, in `table_encourage_per_regularizer.png` within the supplementary material) we can observe that in general, the variance regularizer does no better than others at improving the performance of large models, despite the existence of individual outlier ensembles which perform better. We will include an analysis of which regularizers that we test appear to be the most helpful or harmful overall.
>
> > Reporting only CE loss can limit the explanatory power of the experiments, as one-hot true label is an incorrect categorical distribution for an example. In my opinion, reporting of Accuracy lead to deeper understanding and lead to better insights in what is going on (e.g. provided in Figures 12-20 vs Figures 21-23)
>
> For Figures 21-23 (as well as Figures 3 and 5) we believe that CE loss is the appropriate measure, as it leads to an ensemble loss decomposition (Eqn 3) that allows us to directly compare the contribution of ensemble diversity vs. single model performance to the overall performance of the ensemble. Importantly this decomposition provides a mathematical basis for our claim that ensemble diversity is contributing less to the performance of larger, more powerful component models. Furthermore, NLL is a useful metric in its own right, and is often reported in studies of deep ensembles, where there is a greater focus on probabilistic metrics of performance (e.g. Lakshminarayanan et al. 2016, Jeffares et al. 2023). Unfortunately, no corresponding decomposition exists for predictive accuracy- a point that we can make more clear in the text. However we can certainly add plots which consider the average accuracy of component models relative to some measure of ensemble diversity in the appendix as well.
>
> (Continued)
>
> ### References
> Lakshminarayanan, Balaji, Alexander Pritzel, and Charles Blundell. "Simple and scalable predictive uncertainty estimation using deep ensembles." Advances in neural information processing systems 30 (2017).
>
> Jeffares, Alan, et al. "Joint Training of Deep Ensembles Fails Due to Learner Collusion." arXiv preprint arXiv:2301.11323(2023).

---

> > ### Author Response · Authors · 2023-11-16
> > **(Continued)**
> >
> > > The training procedure can harm the conclusions: simultaneous training of models in an ensemble often leads to the performance degradation on average. The models should be trained indendently with little intersection over losses for them (e.g. boosting-style training).
> >
> > We believe there are several lines of evidence within our work that would suggest that the training procedure is not harming our conclusions. First (as reported in Appendix C.2.1), in Figure 2, the blue bands represent the SEM across independently trained ensemble members. We see that relative to independent training, ensemble models trained with $\gamma = 1$ achieve comparable performance, as expected, since they are training mathematically the same loss. If our training procedure was harming our conclusions, we would already expect to observe a difference between these forms of training. Furthermore, it is not exclusively the case that simultaneous training of models always leads to worse performance, as encouraging diversity on ensembles of small component models can improve performance.
> >
> > Finally, we also demonstrate that the relationship between model capacity and ensemble variance holds on ensembles of independently trained Random Fourier Feature models (Figure 6). Since such models are convex with regards to their parameters, we would expect that the results we see are not specific to any training methodology of neural networks, but are a general phenomenon of large, overparametrized models. We can make this point clearer in the discussion.
> >
> > > Figure 17 y-axis label is wrong
> >
> > Thank you very much for bringing this to our attention- we will be sure to correct this label to read ECE instead of Accuracy.

---

### Review · Reviewer_5PH2 · 2023-10-21

**Summary Of Contributions:**

This paper studies the impact of predictive diversity on the performance of deep neural networks ensembles. It mainly focuses on convolutional neural networks and multiclass classification, while studying a bit also regression (in an appendix). If my understanding is correct, the main finding (claim) of the paper is that in the case of classification a higher predictive diversity helps deep neural network ensembles performance when each ensemble member is a very light model with a small capacity; while when each ensemble model is a very large model with high capacity the predictive diversity hurts the ensemble performance.

**Audience:**

Yes

**Claims And Evidence:**

Yes

**Requested Changes:**

Important aspects:
* 1) Please either change some paper statements to identify well the paper boundaries (e.g., focus and claims empirically demonstrated just on convolutional neural networks and multiclass classification for computer vision), either expand the experimental design to cover at least two extra different Deep Learning architectures and data types (e.g., MLP on tabular data, and whatever else the authors prefer).
* 2) On page 9, the statement about regression “As a first test of these claims, we would predict that regression…” may not be fully correct (I am not sure) and can be an outcome of a flawed experimental design. Can you try the following systematic experiment:
(i) pick a few tabular datasets for regression;
(ii) train simple MLPs with few hidden layers on them. Increase gradually the number of hidden neurons from the hidden layers (e.g., start with a very small number like 5 neurons/layer and go up to ridiculously large numbers for those datasets) and on all these cases study the relation between predictive diversity and (ensemble) models’ performance;
(iii) What is the relation now between predictive diversity and model capacity? Is it still not confirming your findings from the classification experiments?
Alternatively, please consider to remove all statements about regression from the main paper and their corresponding experiments  from the appendix  (anyway, they don’t bring much value in their actual form to the main paper claims) and to make it clear from the begin that the paper focus just on classification.

Minor aspects:
* The last phrase of the abstract “These results call into question the benefit of efforts to create more diverse deep ensembles, especially in the face of an easier alternative: simply forming ensembles from ever more powerful (and less diverse) component models.” is too strong and prunes some valid research directions. Try to ponder down and rephrase a bit this phrase and similar phrases from the main paper to be more in line with your findings and to encourage constructive fundamental research. For instance, just to understand how it can negatively affect the reader and I will give you my case example. After reading the abstract, I had the feeling of a paper which just promotes extremely large networks (and huge amount of compute powers) and somehow I perceived it on the reject side. After reading the paper, I changed my mind, although I still believe that the paper needs a major revision (see my other comments).

**Strengths And Weaknesses:**

Strengths
* The paper is decently well written and relatively easy to read
* The main claim seems novel (I may be wrong as I am not an expert on ensembles and deep learning).
* After reading the paper, I would actually say that the main claim is intuitive and would represent a commonsense perspective. So, it all makes sense to me. A bit surprised that no one did this before when there is such a large body of work on deep ensembles. Can the authors perform a thorough literature survey on this in order to make sure that right credits are given?
* The design of the empirical validation is fine, but not optimal.
* The developed code is provided for easy reproducibility.

Weaknesses
* The paper presents facts, draws some informative conclusions, but it doesn’t explain ‘Why?’. Anyway, answering ‘why’ can be considered (as also the authors mentioned), the focus of follow up works.
*  Several statements in the abstract and the main paper leads the reader to the misleading idea that the paper studied all deep learning architectures which is not true. A common mistake in the literature, but the reality is that Deep Learning is not equal with Convolutional Neural Networks.

---

> ### Author Response · Authors · 2023-11-16
> **Thank you for your reviews!**
>
> Thank you for your evaluation of our paper! We found your suggestions very useful, and we hope that you will find the new results that we present as a result to be sufficient to address them.
>
> > After reading the paper, I would actually say that the main claim is intuitive and would represent a commonsense perspective. So, it all makes sense to me. A bit surprised that no one did this before when there is such a large body of work on deep ensembles. Can the authors perform a thorough literature survey on this in order to make sure that right credits are given?
>
> We are glad that you found the main claims of the paper to be intuitive. While we were also initially surprised by the simplicity of the explanation we discovered, we believe that we have been quite thorough in our survey of the literature between the related work section in the main text and the more detailed comparison in Appendix D (which we can move to the main text if you believe it would be useful). Furthermore, the fact that there are still many papers which propose novel techniques to encourage predictive diversity between ensemble members during training (we reference 5+ papers from 2022 alone) suggests that the claims of our paper are not widely known in the deep ensembling community.
>
> > Please either change some paper statements to identify well the paper boundaries (e.g., focus and claims empirically demonstrated just on convolutional neural networks and multiclass classification for computer vision), either expand the experimental design to cover at least two extra different Deep Learning architectures and data types (e.g., MLP on tabular data, and whatever else the authors prefer).
>
> In response to this suggestion, we have run additional experiments which study the effect of diversity regularization using the Jensen Gap regularizer on the ForestCover tabular dataset, and report accuracy in addition to NLL and Brier score. We consider 4 MLP models, which we identify as **smaller** (2 layers x 32 neurons per layer), **small** (2 layers x 64 neurons per layer) **big** (8 layers x 512 neurons per layer) and **bigger** (8 layers x 1024 neurons per layer). The results of these experiments are currently included in the supplementary material, in the images `kl_weight_forest_mlp_{smaller,small,big,bigger}.pdf`, respectively. Conventions in these figures follow Figure 2.
>
> Importantly, we find that the **smaller** and **bigger** ensembles are entirely consistent with our findings using convolutional networks on vision datasets. On **smaller**, diversity encouragement leads to improvements in model performance across all tested regularization strengths, as we saw in Figure 2. Meanwhile diversity discouragement leads to high variability in accuracy, but overall degradation in NLL and Brier Score. On the **bigger** ensemble, we see that diversity encouragement leads to strictly worse performance across all metrics. Diversity discouragement leads to some degradation in ensemble performance as well.
>
> Interestingly, the ensembles of intermediate size models (**small** and **big**) appear to interpolate between these results. If we look at the **big** ensemble, we find that diversity encouragement first leads to a local maximum in ensemble performance, before declining for all metrics at higher regularization strength, as with **bigger**. Diversity discouragement is also benign at lower regularization values. In contrast, **small** models appear to have high variance in predictive accuracy under diversity encouragement, although NLL and Brier score of these models appears to decline to a local minimum, and then improve at higher regularization strengths, as with **smaller**.
>
> These trends with MLP models suggest not only that our results generalize to other architectures and datasets, but also that it may be interesting to study in future work how the dynamics of how diversity encouragement/discouragement change as we continuously vary component model capacity between the extremes of “large” and “small” models that we focus on. Notably, having finer grained control over MLP architectures allowed us to discover local optima in regularization strength as we consider models of intermediate size, and understanding how these optima evolve along the spectrum of model size will be an important point of future work. We are looking forwards to including these points in our paper.
>
> Although we were unable to consider another dataset type and model architecture due to lack of time, we would like to point out that even the results currently integrated into our paper are not entirely restricted to convolutional neural networks. We demonstrate in Figure 6 that the relationship between ensemble diversity and component model performance is mediated by ensemble size in ensembles of random feature models as well as convolutional networks.
>
> (continued)

---

> > ### Author Response · Authors · 2023-11-16
> > **(Continued)**
> >
> > > On page 9, the statement about regression “As a first test of these claims, we would predict that regression…” may not be fully correct (I am not sure) and can be an outcome of a flawed experimental design. Can you try the following systematic experiment: (i) pick a few tabular datasets for regression; (ii) train simple MLPs with few hidden layers on them. Increase gradually the number of hidden neurons from the hidden layers (e.g., start with a very small number like 5 neurons/layer and go up to ridiculously large numbers for those datasets) and on all these cases study the relation between predictive diversity and (ensemble) models’ performance; (iii) What is the relation now between predictive diversity and model capacity? Is it still not confirming your findings from the classification experiments? Alternatively, please consider to remove all statements about regression from the main paper and their corresponding experiments from the appendix (anyway, they don’t bring much value in their actual form to the main paper claims) and to make it clear from the begin that the paper focus just on classification.
> >
> > We are not sure what potential issue you observe in our experimental design. The model which we use to test our regression experiments is one of the largest that we consider, certainly far larger than others which we also apply to the CIFAR10 and CIFAR100 datasets in a classification context. We are happy to make clear that the main focus of our paper is classification throughout the introduction and setup of our paper. However, we disagree with the suggestion that the experiments in their current form do not contribute to the main claims of the paper. The mechanism that we suggest for why encouraging predictive diversity among ensemble members depends upon the structure of the output space (section 4.2). Regression (specifically classification as regression) provides a setting in which we can consider the same class of models, trained on the same data, and consider the impact of this altered output space. We believe this to be an important test of the mechanism that we propose.
> >
> > > The last phrase of the abstract “These results call into question the benefit of efforts to create more diverse deep ensembles, especially in the face of an easier alternative: simply forming ensembles from ever more powerful (and less diverse) component models.” is too strong and prunes some valid research directions. Try to ponder down and rephrase a bit this phrase and similar phrases from the main paper to be more in line with your findings and to encourage constructive fundamental research. For instance, just to understand how it can negatively affect the reader and I will give you my case example. After reading the abstract, I had the feeling of a paper which just promotes extremely large networks (and huge amount of compute powers) and somehow I perceived it on the reject side. After reading the paper, I changed my mind, although I still believe that the paper needs a major revision (see my other comments).
> >
> > From the results of this paper, we find that there are at least two significant challenges that must be resolved in order to form ensembles that perform better due to increased predictive diversity:
> > - We discuss one of these challenges in section 4.2. When working in bounded output spaces, successfully encouraging predictive diversity would require methods which can encourage predictive diversity on a specific subset of the test data: in particular, on predictions which would be incorrect in the absence of diversity encouragement (i.e. counterfactual errors).
> > - Another challenge is to find sources of diversity between independently trained models that offers a greater advantage than simply using more powerful, and therefore less diverse models. We consider to be a significant challenge given the wide range of model architectures we considered when forming ensembles for this study.
> >
> > We can make sure to qualify our claims in the abstract and main text in two ways: 1) to emphasize that these results primarily apply to a classification context (as mentioned in response to the comment directly above), and 2) by clarifying that it is necessary to overcome the two challenges described above in order to achieve better deep ensembles which are also more diverse. We hope that such revisions would address your concerns that we are pruning valid research directions- please let us know if this is not the case.

---

> > > ### Comment · Reviewer_5PH2 · 2023-12-03
> > > **Thanks for your answers**
> > >
> > > I thank the authors for considering my comments. If their answers (as presented in this discussion) will be integrated in the paper, from my side, I believe that the paper can be accepted.
> > >
> > > About the regression comment, I believe that my remark was a bit misunderstood. I agree that the model used for regression in the paper is large, but this was not my concern. I believe that what is missing with respect to regression is a systematic approach where the model size is gradually increased (from very small to very large). Anyway, if the authors will mention in the revised version that the paper focus is classification, this experiment is not necessary and can be the subject of a follow up work.

---

### Review · Reviewer_dz3x · 2023-11-02

**Summary Of Contributions:**

This work provides an empirical analysis of the role of predictive diversity in deep ensembles. Therefore, the primary contributions lie in the results of the investigation of the research questions posed. The first of these is that, when optimizing a deep ensemble, encouraging diversity is beneficial for small networks but not for larger networks and, relatedly, discouraging diversity has minimal effect on large networks but a harmful effect on smaller networks. This appears to be related to the fact that larger models tend to obtain very high accuracy and, as a result, diversity can only come at the expense of predictive performance on individually strong models. Next, the authors investigate if increasing diversity _implicitly_ (e.g. via diverse architectures) can still benefit from diversity. Even in this case, it appears that the marginal gains from diversity are still less performant than simply ensembling with individually strong but low diversity ensembles. In addition to these results, some minor contributions are provided including an analysis of the ensemble objective and the choices of diversity measure as well as a helpful overview of the existing work on optimizing for diversity including Table 2 which highlights that previous works have generally failed to obtain performance gains by optimizing for diversity on the training set.

**Audience:**

Yes

**Claims And Evidence:**

Yes

**Requested Changes:**

I request that the figures mentioned be tweaked to improve their clarity. I have provided some suggestions on what might help, but I leave it to the authors to implement whatever changes they find to work best (i.e. I am not insisting on these specific suggestions).

* Fig 1 - Even this relatively simple opening figure is a little tricky to "get".  Why is the blue/red line two colors, could it not just be blue _or_ red? One option might be an additional arrow showing the directions of loss decreasing (i.e. perpendicular to the level set line). Or some shading maybe?
* Fig 3 - it would be nice to see the actual max and min values of $\gamma$ used in addition. Also, the text is too small. It also seems that the plots could be zoomed in to make it easier to read the points in the plot.
* Fig 4 (left and center left) - I found this to be the most challenging plot to understand of all (although the result it conveys is important once understood). Some factors that contributed to my confusion were: (a) counterfactual accuracy is only defined in the main text, not in the caption or the figure itself. I had no intuition for what this meant; (b) there seems to be more lines/colors in the plot than in the legend; (c) lines for density plots are a little non-standard; (d) it took me a while to notice what was the difference between the left and center-left plots (i.e. ResNet-8 vs ResNet-18).
* Fig 5 - Again, seems to be overly zoomed out. xlims and ylims could be reduced.

Finally, I have some additional suggestions.
* Although it is not the main point of the paper, I think it would be useful for future research to include some discussion comparing the four diversity encouraging/discouraging mechanisms from page 5 (even in the appendix). Given the authors have run considerable experiments with these terms, some high-level perspective on their relative strengths/weaknesses might be useful for future research that intends to optimize for diversity (where it is suitable).
* As mentioned in my questions, I think a summary of the paper's contributions in the introduction would help anchor the reader in the text.

**Strengths And Weaknesses:**

**Strengths**

This work addressed some challenging questions that I believe to be of substantial interest to the community. Overall, I am very positive about this work and found it to be a focused but rigorous analysis.

* Given the broad application of ensembling to boost the performance of classifiers in practice I think this paper addresses an important area. I think the conclusions of this work are relatively counter to the general prevailing wisdom on how deep ensembles should be constructed and trained. Therefore, I think this work provides a significant contribution.
* The main result in my view is that model capacity is a determining factor in the value of diversity. While original research in the statistics literature developed theory around ensembles of _weak_ learners, the results in this work provide strong evidence that these intuitions do not carry over to the high capacity deep ensemble regime. This is a valuable finding.
* The explanation of the cause of the issue (i.e. highly accurate predictions having diversity enforced causes these predictions to become inaccurate) was intuitive and well supported. I found this explanation convincing.
* I think that this work should both (a) spark new research (e.g. developing theory around these empirical findings, investigating new approaches beyond the diversity perspective for improving deep ensemble performance) and (b) direct research away from the ineffective approach of optimizing for train set diversity.
* In general, I thought the experiments were comprehensive and well-designed (although I have some minor suggestions on their presentation in some figures).
* The paper was generally well-written and clear.
* Many additional results are provided in the appendix which reinforces the points made in the main text. Code is also provided.


**Weaknesses**

I did not find any major weaknesses in this work. As a minor note, I thought that some of the figures could have been more intuitive. At times it took me quite a while to fully understand what was being shown (I provide some specific suggestions in requested changes). I would suggest that it might also be useful to provide a summary of the key takeaways in a list or a table up front. Since there is quite a lot to process in throughout the paper it might be nice to have something to anchor the reader to where they are in the paper.

**Questions**

I have some questions that don't fit as strengths or weaknesses.
* Are the results provided in Fig 8 on the validation set? I presume so, but I don't think it is explicitly stated.
* Notation: Is there a reason $\lambda$ is used in Table 1 for regularization strength rather than $\gamma$?
* One thing that is a little unclear is if it is model capacity or individual model performance that dictates whether we are in the _weak learner_ regime or the _strong learner_ regime. I suspect it to be the latter but, for a fixed dataset, increasing capacity results in improved model performance. I suspect that this might be a useful point to be clear on for future work that might analyze this change of regime from a model complexity perspective. Do the authors agree with me on this? If so, it might be worth mentioning this point somewhere in the text.

---

> ### Author Response · Authors · 2023-11-16
> **Thank you.**
>
> Thank you very much for your thoughtful evaluation of our work! We are very appreciative of the feedback that you have provided, and we have described below the steps that we will take to address your comments, especially around improving the presentation of our results.
>
> First, we will aim to answer the questions you posed:
>
> > Are the results provided in Fig 8 on the validation set? I presume so, but I don't think it is explicitly stated.
>
> Yes- we will be sure to clarify this in the caption to Figure 8, as well as Appendix Section E.2.
>
> > Notation: Is there a reason $\gamma$ is used in Table 1 for regularization strength rather than $\lambda$?
>
> Thank you for catching this- this is an error on our part, and we will fix the table to parametrize the regularization strength by $\gamma$, as in the rest of our paper.
>
> > One thing that is a little unclear is if it is model capacity or individual model performance that dictates whether we are in the weak learner regime or the strong learner regime. I suspect it to be the latter but, for a fixed dataset, increasing capacity results in improved model performance. I suspect that this might be a useful point to be clear on for future work that might analyze this change of regime from a model complexity perspective. Do the authors agree with me on this? If so, it might be worth mentioning this point somewhere in the text.
>
> This is a very interesting point. Fundamentally, we expect that the difference between these two regimes is due to the proportion and quality (correct vs. incorrect) of predictions that are made with high confidence.  While both the performance of individual models model capacity and has been correlated with high confidence predictions in previous research (e.g. Guo et al. 2017, Nakkiran et al. 2019, Bernstein et al. 2022), as you state it is unclear which of these factors dictates weak learner vs. strong learner behavior. We will add a section to our discussion which explicitly discusses this fact. We will also relate this discussion back to our results in Figure 6, where we see different forms of this transition from the weak learner to the strong learner regime between random fourier feature models and various families of deep neural networks.
>
> Next, we will aim to address your requested changes:
>
> > Fig 1 - Even this relatively simple opening figure is a little tricky to "get". Why is the blue/red line two colors, could it not just be blue or red? One option might be an additional arrow showing the directions of loss decreasing (i.e. perpendicular to the level set line). Or some shading maybe?
>
> We chose to make the ensemble level set blue on the left, and red on the right of ensemble performance in order to visually distinguish strategies that improve ensemble performance by encouraging predictive diversity or sacrificing it in favor of ensemble performance. We agree that this clutters the figure, and we are happy to replace these lines with a single thicker line accompanied by a more thorough explanation in the main text and caption.
>
> >Fig 3 - it would be nice to see the actual max and min values of γ used in addition. Also, the text is too small. It also seems that the plots could be zoomed in to make it easier to read the points in the plot.
>
> We can certainly reformat these figures in order to better fill the graphs, increase text size, and add in the explicit range of gamma values used. Thank you for the suggestion.
>
> (continued in next comment)
>
> ### References
> Guo, Chuan, et al. "On calibration of modern neural networks." International conference on machine learning. PMLR, 2017.
>
> Nakkiran, Preetum, et al. "Deep double descent: Where bigger models and more data hurt." Journal of Statistical Mechanics: Theory and Experiment 2021.12 (2021): 124003.
>
> Bernstein, Jeremy, Alex Farhang, and Yisong Yue. "Kernel Interpolation as a Bayes Point Machine." arXiv preprint arXiv:2110.04274 (2021).

---

> > ### Author Response · Authors · 2023-11-16
> > **(Continued)**
> >
> > > Fig 4 (left and center left) - I found this to be the most challenging plot to understand of all (although the result it conveys is important once understood). Some factors that contributed to my confusion were: (a) counterfactual accuracy is only defined in the main text, not in the caption or the figure itself. I had no intuition for what this meant; (b) there seems to be more lines/colors in the plot than in the legend; (c) lines for density plots are a little non-standard; (d) it took me a while to notice what was the difference between the left and center-left plots (i.e. ResNet-8 vs ResNet-18).
> >
> > Thank you for this feedback. We will:
> > - a) ensure that we include a definition of counterfactual accuracy in the text caption, and likewise provide more intuition for what this quantity represents in addition to the definitions given (i.e., we will state that it describes the accuracy we would expect from an ensemble in the absence of a diversity intervention.)
> > - b) We will also provide a legend that lists all lines/colors used in the plot.
> > - c) We are in favor of lines in this case in order to more compactly represent density information across multiple different regularization strengths. However, we will attempt to represent this information with more traditional shaded densities, as well as other more standard alternatives.
> > - d) We will indicate the difference between these plots through larger titles.
> >
> > > Fig 5 - Again, seems to be overly zoomed out. xlims and ylims could be reduced.
> >
> > Thank you for these suggestions: we will crop out empty whitespace in these plots.
> >
> > > Although it is not the main point of the paper, I think it would be useful for future research to include some discussion comparing the four diversity encouraging/discouraging mechanisms from page 5 (even in the appendix). Given the authors have run considerable experiments with these terms, some high-level perspective on their relative strengths/weaknesses might be useful for future research that intends to optimize for diversity (where it is suitable).
> >
> > We agree that this would be useful for future research. We will add an appendix section that compares these mechanisms further. In particular, we will be sure to discuss differences in:
> > - The effective ranges of these regularizers: some are unbounded, others have limits in regularization strength beyond which they are expected to diverge.
> > - The portions of the predictions which they aim to diversify: some target diversity in the entire probability vector, while others target diversity only in the true class label.
> > - The empirical effectiveness of each diversity regularizer. We include a breakdown of our overall quantification of the effect of diversity regularization per regularizer, currently in the supplementary material. The file `table_encourage_per_regularizer.png` breaks down the performance of each diversity regularizer on all instances of diversity encouragement in the models we trained, and likewise `table_discourage_per_regularizer.png` breaks down performance for all regularizers where we discouraged predictive diversity. These plots reveal interesting trends in the overall effectiveness (or lack thereof) of each regularizer in different conditions, as well as identifying which regularizers are most or least harmful to performance across the range of tested conditions.
> >
> > > As mentioned in my questions, I think a summary of the paper's contributions in the introduction would help anchor the reader in the text.
> >
> > Thank you for this point. We will add a summary of our main contributions to the beginning of the paper, as follows:
> >
> > We demonstrate that the size of component models determines the role of predictive diversity in ensemble performance. We show this in two sets of experiments:
> > 1. We show that optimization objectives that can encourage or discourage predictive diversity between ensemble members succeed in increasing predictive accuracy on ensembles of small models, but these same objectives fail when applied to ensembles of large component models. We demonstrate this result on over 600 deep ensembles, across various architectures, datasets, and training objectives, and show that our findings reconcile differences between the traditional ensembling literature with methods proposed to improve deep ensembles.
> > 2. Additionally, we show that our results generalize to many forms of implicit diversity encouragement, i.e. by ensembling across different neural network architectures and hyperparameters. We find that while there may be marginal gains from ensembling more diverse models in such a context, they are vastly outweighed by the improvements that can be gained by simply ensembling better, less diverse component models.

---

> > > ### Comment · Reviewer_dz3x · 2023-11-27
> > > **Thank you**
> > >
> > > I am grateful to the authors for their comprehensive response. I am confident that this paper is a clear accept and provides a substantial contribution to the community.

---

### Author Response · Authors · 2023-11-16
**Thank you for your reviews!**

We would like to thank the reviewers for their insightful comments, which we have considered carefully and are using to significantly update the content of our paper as described in the responses below. We are currently revising our paper, but have indicated the content of any meaningful revisions in our responses below. In addition, we have added new figures, currently uploaded as part of the supplementary material that we will reference throughout our responses. We will integrate these figures into a full, updated revision as soon as possible.

---

### Author Response · Authors · 2024-01-02
**Thank you!**

We have posted a camera-ready version of our manuscript which integrates our new results, and addresses all additional points of reviewer feedback around the framing, presentation, and interpretation of our findings. We believe that our paper has been greatly improved as a function of the review process, and we would once again like to thank the reviewers and the action editor for their feedback and discussion of our work!

---

### Decision · Action_Editor_7Ure · 2023-12-14

**Recommendation:** Accept with minor revision

**Comment:**

All reviewers are supportive of this paper, and it clearly meets the bar for acceptance.

"Minor revision" selected because the authors still need to integrate the figures provided during the rebuttal into the main pdf, and make the small changes promised in the responses. Other than those items, no changes needed.

**Audience:**

Ensembling is a long-standing and well-known ML technique, and continues to be studied in the context of deep learning. Diversity-promoting methods are still a popular area of study. This paper provides clear conclusions with a strong empirical study that will likely be useful to both researchers in the field and practioners alike.

**Claims And Evidence:**

The reviewers are unanimous in their agreement that the claims of the paper are well supported by the experimental evidence in the paper. Only a few weaknesses were surfaced during the reviews, and these were well addressed. The additional controlled study on MLP-based classification of tabular data is appreciated, and it further supports the findings.